# Compression-induced expression of glycolysis genes in CAFs correlates with EMT and angiogenesis gene expression in breast cancer

Baek Gil Kim [1,2], Jin Sol Sung[2], Yeonsue Jang[1], Yoon Jin Cha[1], Suki Kang[1,3], Hyun Ho Han[2], Joo Hyun Lee[2] & Nam Hoon Cho[1,2,3,4]

Tumor growth increases compressive stress within a tissue, which is associated with solid tumor progression. However, very little is known about how compressive stress contributes to tumor progression. Here, we show that compressive stress induces glycolysis in human breast cancer associated fibroblast (CAF) cells and thereby contributes to the expression of epithelial to mesenchymal (EMT)- and angiogenesis-related genes in breast cancer cells. Lactate production was increased in compressed CAF cells, in a manner dependent on the expression of metabolic genes *ENO2*, *HK2*, and *PFKFB3*. Conditioned medium from compressed CAFs promoted the proliferation of breast cancer cells and the expression of EMT and/or angiogenesis-related genes. In patient tissues with high compressive stress, the expression of compression-induced metabolic genes was significantly and positively correlated with EMT and/or angiogenesis-related gene expression and metastasis size. These findings illustrate a mechanotransduction pathway involving stromal glycolysis that may be relevant also for other solid tumours.

[1] Department of Pathology, Yonsei University College of Medicine, Seoul, South Korea. [2] Brain Korea 21 Plus Project for Medical Science, Yonsei University College of Medicine, Seoul, South Korea. [3] Severance Biomedical Science Institute (SBSI), Yonsei University College of Medicine, Seoul, South Korea. [4] Global 5-5-10 System Biology, Yonsei University, Seoul, South Korea. Correspondence and requests for materials should be addressed to N.H.C. (email: cho1988@yuhs.ac)

Biochemical and mechanical changes in tumor microenvironment are both associated with tumor progression. However, the latter is not studied as comprehensively as the former in recent cancer research. Considering that uncontrolled tumor growth inevitably increases mechanical stresses within almost all solid tumor tissues, studying the relationship between mechanotransduction and tumor progression phenotypes may be an important area in cancer research and treatment.

Tumor progression phenotypes may be outcomes from the compression-induced transcriptomic alterations of tumor and stromal cells. Unlimited cell proliferation, a fundamental property of cancer, causes various mechanical stresses within cancer tissues[1,2]. Among them, compressive stress is firstly and directly generated by the proliferation of tumor cells within a limited tissue space. Stylianopoulos et al. showed that tumor growth increases compressive stress in the interior and periphery of tumor tissue[3]. In the tumor periphery, compressive stress can induce tumor progression phenotypes. Tse et al. demonstrated that compressive stress stimulates the formation of leader cells at the edge (periphery) of the cell sheet[4]. We previously found that gene expression alteration in the interface zone (IZ), the fibrotic tissue 10 mm away from the periphery of tumor tissue, was associated with breast cancer invasiveness. Some proteins known to contribute to tumor progression like LAMC2, ITGA6, and ITGB4 are overexpressed in the IZ of invasive ductal carcinoma (IDC) compared to the counterpart of ductal carcinoma in situ[5]. It was previously demonstrated using an in vitro compression model that compressive stress induces the overexpression of LAMC2 and ITGA6 in some breast cancer and cancer-associated fibroblast (CAF) cells, which lead to the production of VEGFA, a proangiogenic factor, in CAF cells[6]. Thus, tumor progression phenotypes can be induced in the tumor and stromal cells by compression-induced transcriptomic alterations, specifically located around the tumor periphery.

In this study, we first analyzed the compression-induced transcriptomic alterations of breast cancer cells and CAF cells in order to find potential biological processes, which are not only induced by compressive stress but also associated with tumor progression. We then demonstrated how compressive stress induces biological processes using functional assays and further validated our findings using patient tissues with a high compressive stress and a cancer database.

## Results

**An in vitro compression model using alginate disks to study compression-induced mechanotransduction in tumor and stromal cells.** Tumor growth increases compressive stress in the IZ (tumor periphery) between tumor and stromal tissues (Fig. 1a)) why mechanotransduction in this area may be critical for tumor progression. To extract RNAs and proteins from compressed cells, we established an in vitro compression model using alginate disk (Fig. 1b). The transfer of compressive stress to the cells embedded in an alginate disk was indirectly confirmed by measuring the disk's thickness. In Fig. 1c, alginate disk thickness was decreased proportionally to the degree of compressive stress (0–7.732 kPa, 0.773 kPa—the compressive stress value of a native tumor microenvironment[4]). However, 7.732 kPa of compressive stress frequently broke alginate disks. We next examined whether cell viability is associated with the degree of compressive stress. As shown in Fig. 1d, cell viability was not significantly different from 0 to 3.866 kPa, but it was significantly decreased at 7.732 kPa. The alginate disk was suitable to contain cells under compressive conditions. In Fig. 1e, most cells (RFP-positive) were within an alginate disk under compression (3.866 kPa for 1 day). Alginate disk deformation by compressive stress

may cause the limitation of oxygen and nutrients, which affects gene expression in cells. Therefore, the diffusion rate of Ponceau S (a red-colored dye having a molecular weight of 750 Da), which is a larger molecule than all medium components (less than 500 Da), was measured from the alginate disks exposed to different degrees of compressive stress. However, the degree of compressive stress tested did not affect the diffusion rate of Ponceau S on the alginate disks (Supplementary Fig. 1).

**The functional enrichment analysis of compression-induced transcriptomic alteration.** To investigate whether compressive stress induces the biological processes (BPs) possibly related to tumor progression, we sorted the genes commonly being over 2-fold upregulated or downregulated at all compressive conditions (0.386, 0.773, 1.546, 3.866, and 7.732 kPa) from the transcriptome profiling data of breast cancer cell lines (BT-474: luminal B, MCF7: luminal A, SK-BR-3: HER2, and MDA-MB-231: triple-negative) and four patient-derived CAF cells (from invasive ductal carcinoma, stage 1) and then analyzed them using The Database for Annotation, Visualization, and Integrated Discovery (DAVID) Bioinformatics Resources 6.7[7]. The clustering of BPs was performed with the stringencies of medium, high, and highest. The cutoff enrichment score for significant clustering was 1.3, equivalent to non-log scale 0.05[7]. The functional annotation clustering results are provided as Supplementary Data 2. As shown in Fig. 2, homophilic cell adhesion was enriched from the downregulated genes of BT-474, SK-BR-3, MDA-MB-231, CAF2, and CAF4 cells. On the other hand, metal ion homeostasis was enriched from the upregulation genes of CAF2, CAF3, and CAF4 cells. Glycolysis was enriched from the upregulation genes of MDA-MB-231, CAF2, and CAF4 cells. Regulation of cell migration was enriched from the upregulated genes of SK-BR-3, CAF2, and CAF4 cells. The enrichment score of homophilic adhesion showed an average over 5, whereas those of metal ion homeostasis, glycolysis, and regulation of cell migration were all below 5.

**Glycolysis can be promoted in CAF cells by compression-induced transcriptomic alteration.** For the supervised validation of the functional enrichment clustering, based on the gene ontology from Amigo 2[8,9], we sorted the genes involved in the enriched BPs from the transcriptome profiling data and then analyzed their expression using dot distribution graphs to present both gene expression values at 0.773 kPa and average gene expression values at all compressive conditions. The predominant changes in BT-474 cells were downregulation of genes in homophilic cell adhesion and upregulation of glycolysis (Fig. 3a). MCF7 cells showed downregulation of genes in negative and positive regulation of cell migration and glycolysis. SK-BR-3 showed downregulation of genes in negative regulation of cell migration and upregulation in positive regulation of cell migration. MDA-MB-231 cells showed downregulation of genes in homophilic cell adhesion and negative and positive regulation of cell migration, but upregulation in glycolysis. CAF1 cells showed upregulation of genes in homophilic cell adhesion, negative and positive regulation of cell migration, and glycolysis, whereas CAF2 cells did upregulation in positive regulation of cell migration and glycolysis. CAF3 cells showed downregulation of genes in homophilic cell adhesion and negative and positive regulation of cell migration and upregulation of glycolysis. CAF4 cells also showed a dominant upregulation of genes in glycolysis. To understand the compression-induced BP alteration tendencies between cell types, we compared the upregulated/downregulated gene number ratio in each category of the BPs. As shown in

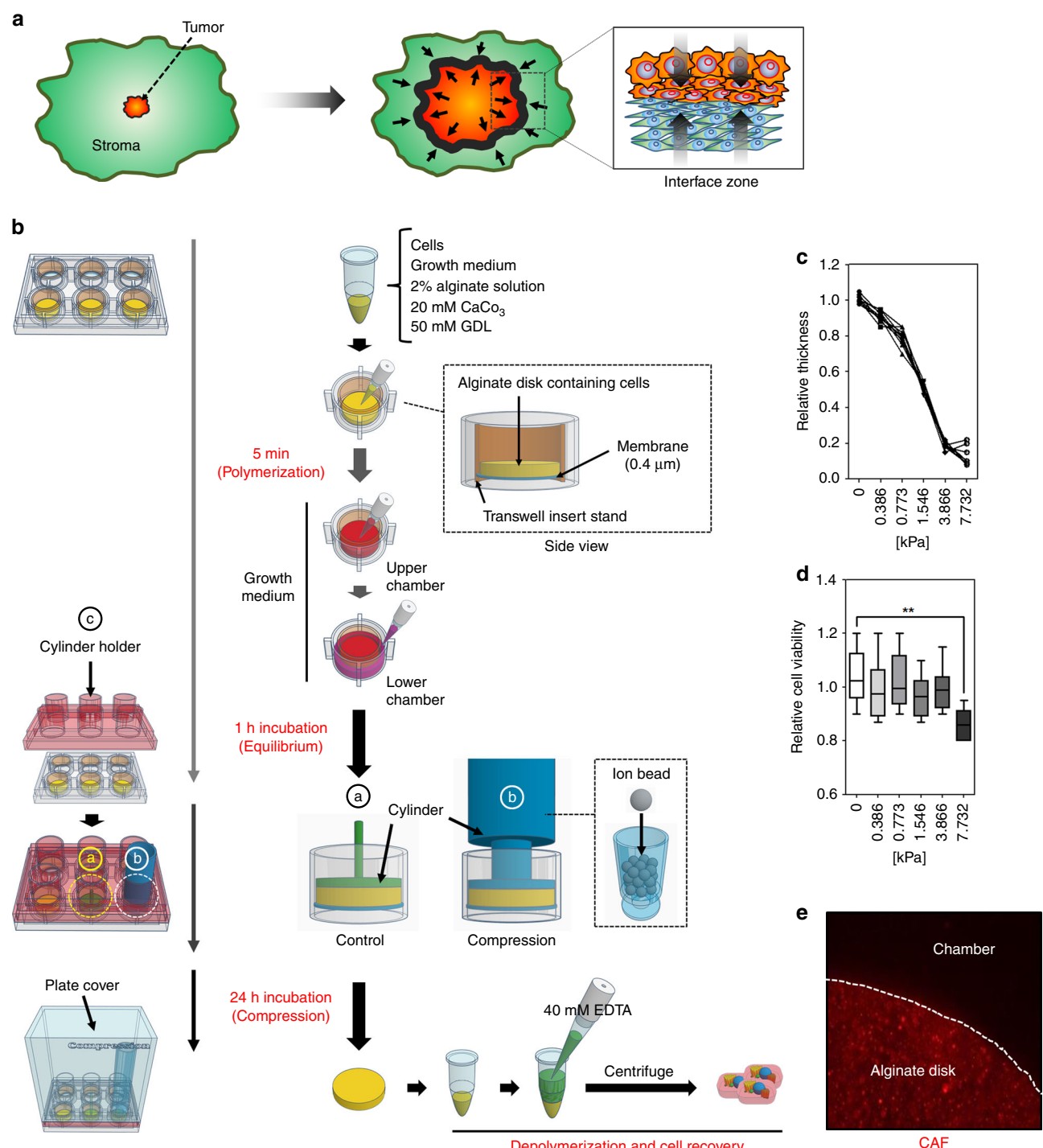

**Fig. 1** An in vitro compression model using alginate disk. **a** Illustration of tumor growth-generated compressive stress in the interface zone. Tumor tissue is surrounded by fibrotic stromal tissue. Therefore, tumor growth increases compressive stress in the interface zone (tumor periphery), and thereby triggers mechanotransduction in both tumor and stromal cells. **b** The schematic process of an in vitro compression model with an alginate disk. The alginate disk containing cells is constructed on the membrane of transwell insert stand. After equilibrium with growth medium for 1 h, the alginate disk is compressed by using the cylinder filled with iron beads and cylinder holder. **c** Compression-dependent deformation of the alginate disk. A cylinder-shaped alginate disks were compressed by loading weight. Alginate disk thickness was measured 24 h after loading weight. **d** The comparative viability of the cells exposed to different degree of compressive stress. The relative viability of the cells was measured from the alginate disks containing cells using cell counting assay ($n = 6$ independent experiments). Error bars and $p$-values were determined by Whiskers (Min to Max) and unpaired two-tailed $t$-test, respectively. **e** The immunofluorescence image of the alginate disk containing cells. The CAF cells labeled with CellTracker Red were embedded into an alginate disk, placed on a well of 6-well tissue culture plate, incubated in growth medium for 1 h, and exposed to 3.866 kPa for 1 day. Source data are provided as Supplementary Data 1

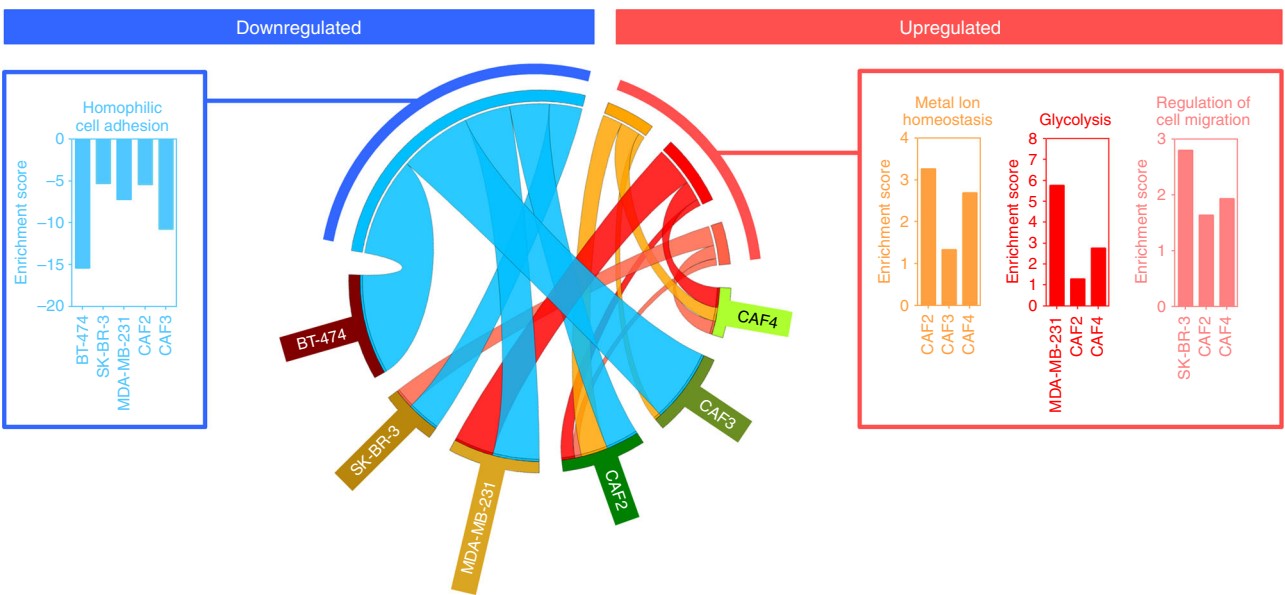

**Fig. 2** The functional annotation clustering of compression-induced transcriptomic alteration. The genes commonly being over 2-fold upregulated or downregulated at all compressive stresses were analyzed by using DAVID. The presented BPs had over 1.3 enrichment score and were found in at least three types of cells. For a direct comparison between cell types and BPs, bar graphs were presented together with Circos display[52]. Source data are provided as Supplementary Data 1

Fig. 3b, glycolysis showed a dominant positive tendency in most types of cells except for MCF7 and SK-BR-3 cells.

**Compressive stress promoted glycolysis in cancer-associated fibroblasts by inducing the upregulation of *ENO2*, *HK2*, and *PFKFB3* genes**. Based on the functional enrichment clustering and its supervised validation, glycolysis is likely to be promoted by compressive stress in MDA-MB-231 and CAF cells. Since compression-induced promotion of glycolysis can be caused by upregulation of related genes, the expression of glycolysis-related genes was first analyzed using heatmap clustering. The *ENO2* (enolase 2), *HK2* (hexokinase 2), and *PFKFB3* (6-Phosphofructo-2-Kinase/Fructose-2,6-Biphosphatase 3) genes were commonly upregulated at all compressive stress conditions in four patient-derived CAF cells (Fig. 4a). On the other hand, those genes were not upregulated in MDA-MB-231 cells. Based on this result, we hypothesized that compressive stress promotes glycolysis in CAF cells via the upregulation of *ENO2*, *HK2*, and *PFKFB3* genes. To understand whether the expression of *ENO2*, *HK2*, and *PFKFB3* genes is proportionally increased to the degree of compressive stress, the signal intensities of those genes were verified from the transcriptome profiling data. As shown in Fig. 4b, the expression of *ENO2*, *HK2*, and *PFKFB3* genes was generally not proportional to the degree of compressive stress. Among the CAF cells, CAF2 cells showed the highest expression of *ENO2*, *HK2*, and *PFKFB3* genes in response to compressive stress. Therefore, CAF2 cells were mainly used for the following experiments as a representative for CAF cells. The compression-induced upregulation of *ENO2*, *HK2*, and *PFKFB3* genes was confirmed in the CAF cells exposed to 0.386 kPa, the compressive stress value of a native tumor microenvironment, using real-time PCR analysis. As shown in Fig. 4c, the expression of ENO2, HK2, and PFKFB3 mRNAs were significantly upregulated by compressive stress in CAF cells. Next, we investigated whether compressive stress can promote the production of lactate, the final product of glycolysis, in CAF cells. As shown in Fig. 4d, the amount of lactate was significantly increased at 0.386, 0.773, 1.546, 3.866, and 7.732 kPa compared to the control (the CAF cells at 0 kPa). To confirm that the compression-induced

promotion of lactate production in CAF cells is dependent on the expression of *ENO2*, *HK2*, and *PFKFB3* genes, the amount of lactate was measured in the CAF cells with or without compression and/or shRNA transfection against corresponding genes. Indeed, the amount of lactate was significantly decreased in CAF cells by the knockdown of *ENO2*, *HK2*, and *PFKFB3* genes in the absence of compression (Fig. 4e and Supplementary Fig. 2). Similarly, in the presence of compression, the amount of lactate was significantly reduced by the knockdown of *ENO2*, *HK2*, and *PFKFB3* genes (Fig. 4f).

**Compressive stress induced the expression of *ENO2*, *HK2*, and *PFKFB3* genes in cancer-associated fibroblasts via c-Jun activation.** c-Jun and c-Fos are activated by mechanical stress[10] and fibrosis is associated with JNK (c-Jun terminal kinase)-induced stress signaling[11]. It is therefore plausible that compressive stress promotes the expression of *ENO2*, *HK2*, and *PFKFB3* genes via the activation of c-Jun and/or c-Fos. To investigate whether there are binding sites for c-Jun or c-Fos on *ENO2*, *HK2*, and *PFKFB3* gene promoters, the 2 kb upstream sequences of the genes were analyzed using TRANSFAC database[12,13]. In the sequence analysis with the maximum matrix dissimilarity rate of 5, the prediction results were as follows: two binding sites (−1282 to −1275, −1151 to −1144) for c-Jun and one binding site (−1281 to −1271) for c-Fos in *ENO2* gene promoter, two binding sites (1687 to −1680, −578 to −571) for c-Jun in *HK2* gene promoter, one binding site (−324 to −317) for c-Jun in *PFKFB3* gene promoter. Since c-Jun was commonly predicted in all three genes, c-Jun binding to those genes were evaluated using dual-luciferase assay. In Fig. 5a, the wild-type (WT) and deletion mutant type (M) of gene constructs were cloned into pGL3 vector. In the luciferase assay of *ENO2* gene promoter, M1 (−1282 to −1275) did not affect on luciferase activity, whereas M2 (−1151 to −1144) induced a significant decrease of luciferase activity compared to WT. In the luciferase assay of *HK2* gene promoter, both M1 (−1687 to −1680) and M2 (−578 to −581) induced a significant decrease of luciferase activity compared to the WT. M1 + M2 showed a similar level of luciferase activity to M1 and M2. In the luciferase assay of *PFKFB3* gene promoter, M (−324 to

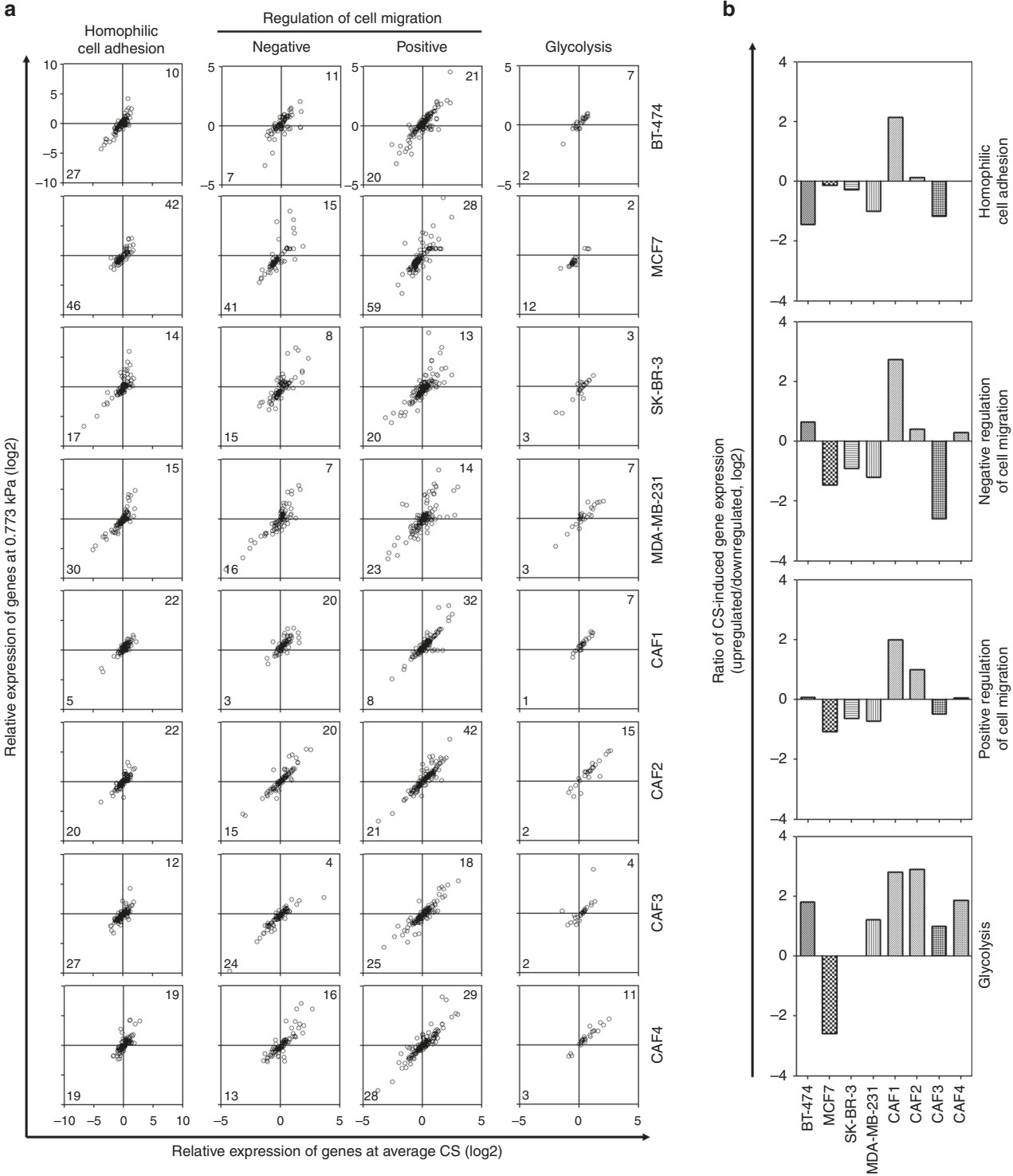

**Fig. 3** Compression-induced expression alteration of enriched biological process-related genes. **a** The expression distribution analysis of the genes involved in homophilic cell adhesion, the negative and positive regulation of cell migration, and glycolysis. Based on the gene ontology from Amigo 2, the genes corresponding to each biological process were classified and analyzed. The $x$- and $y$-axis represent the average values of gene expression at different compressive stress conditions and the values of gene expression at 0.773 kPa, respectively. The genes with the values greater than or equal to 2 both in $x$- and $y$-axes were counted in the upper-right corner of graphs, whereas the genes with less than -2 value in both in $x$- and $y$-axes were counted in the lower-left corner. **b** The ratio presentation of the expression distribution analysis. The ratios were calculated by dividing the number of upregulated genes by that of downregulated genes. CS is the abbreviation of compressive stress. Source data are provided as Supplementary Data 1

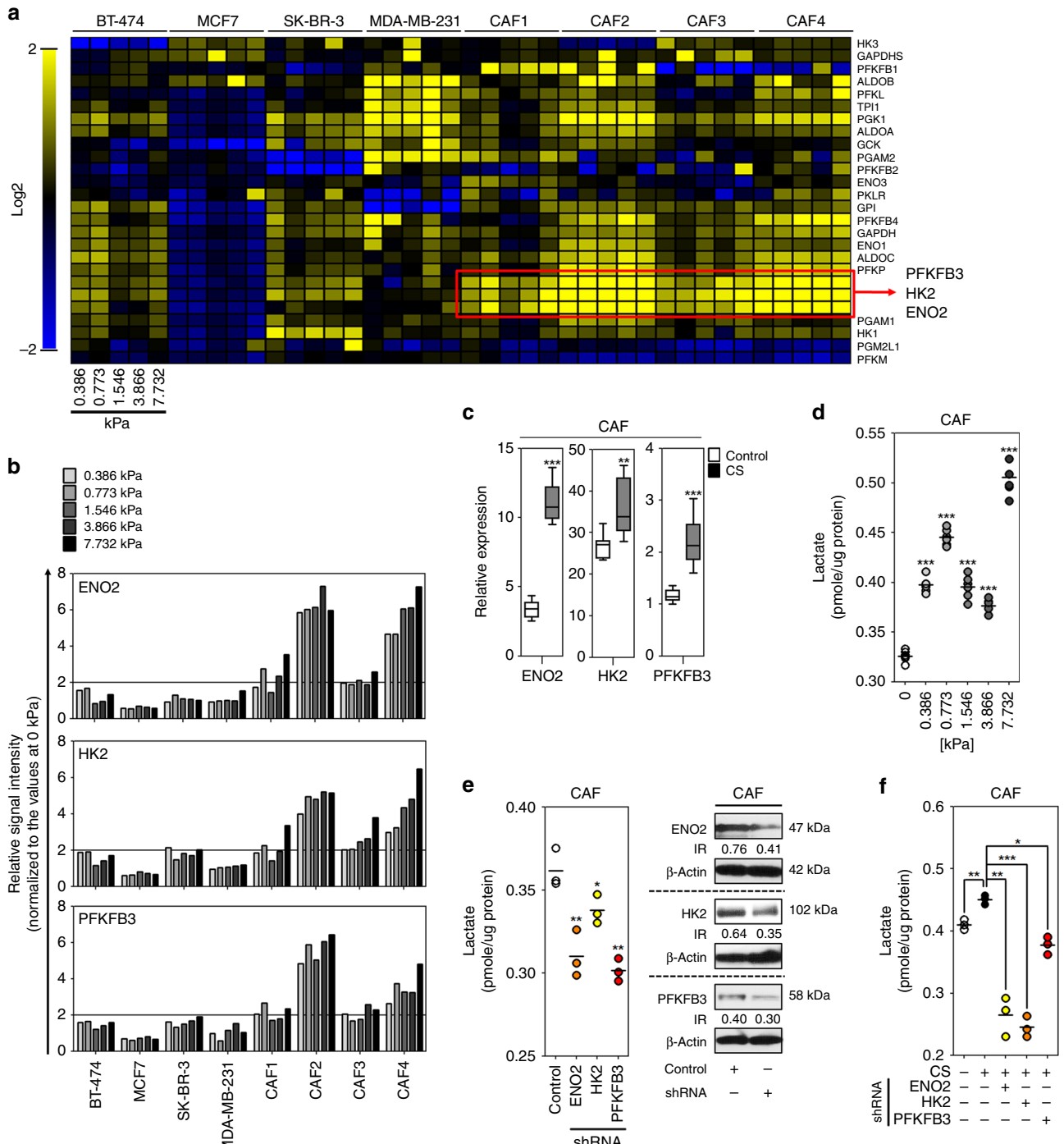

**Fig. 4** The compression-induced promotion of glycolysis in cancer-associated fibroblasts is associated with the upregulation of *ENO2*, *HK2*, and *PFKFB3* genes. **a** The heatmap clustering analysis of glycolysis-related genes. The genes were sorted from the transcriptome profiling data based on Amigo 2 database, and then clustered by Multiple Array Viewer (MeV, version 4.9.0). **b** The relative signal intensities of *ENO2*, *HK2*, and *PFKFB3* genes in the cells exposed to different degrees of compressive stress. The signal intensity values of the genes were obtained from the transcriptome profiling data. **c** The compression-induced upregulation of ENO2, HK2, and PFKFB3 mRNAs. Total RNA was extracted from the CAF cells exposed to 0.773 kPa for 1 day, reverse-transcribed, and then analyzed using real-time PCR ($n = 9$ independent experiments). **d** The compression-induced promotion of lactate production. CAF cells were exposed to the corresponding compressive stress for 1 day ($n = 6$ independent experiments). **e** The decreased production of lactate by gene knockdown. CAF cells were transfected with the shRNAs against *ENO2*, *HK2*, or *PFKFB3* gene ($n = 3$ independent experiments). **f** The effect of gene knockdown on the compression-induced promotion of lactate production. CAF cells were transfected with the shRNAs against *ENO2*, *HK2*, or *PFKFB3* gene, and then exposed to 0.773 kPa ($n = 3$ independent experiments). Error bars and p-values were determined by Whiskers (Min to Max) and unpaired two-tailed *t*-test, respectively. CS and IR are the abbreviation of compressive stress and intensity ratio, respectively. Source data are provided as Supplementary Data 1

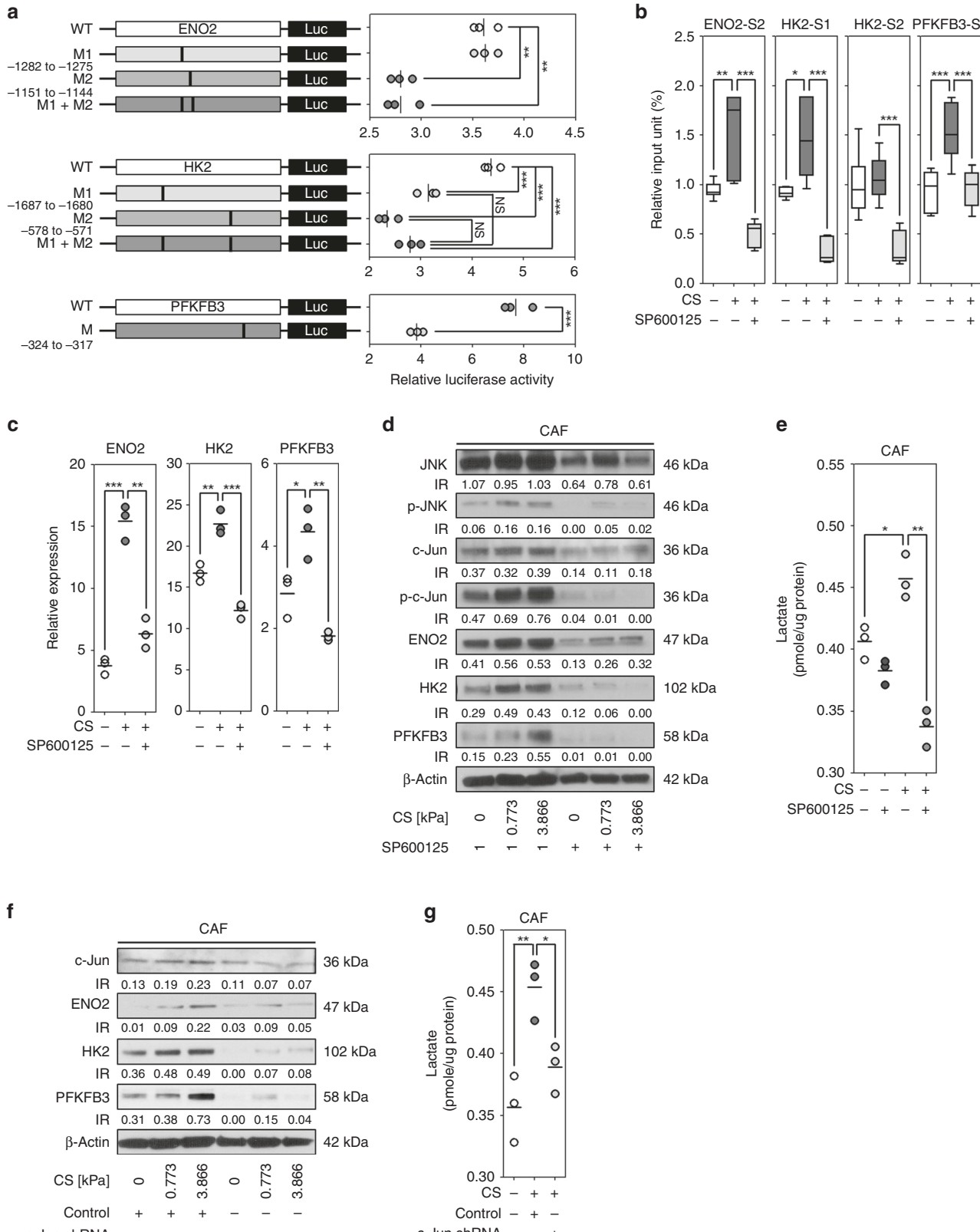

−317) showed a significant decrease of luciferase activity compared to WT. Next, c-Jun binding to the *ENO2*, *HK2*, and *PFKFB3* gene promoters were further evaluated in the CAF cells with or without compressive stress and/or c-Jun inhibitor (SP600125) using chromatin immunoprecipitation. Confirming the results of the luciferase assay, c-Jun binding to the site 2 (S2,

−1151 to −1144) of *ENO2* gene promoter was significantly increased by compression compared to the control, which was inhibited by SP600125 pretreatment. c-Jun binding to the site 1 (S1, −1687 to −1680) of *HK2* gene promoter was significantly increased by compression compared to the control, which was suppressed with the pretreatment of SP600125. c-Jun binding to

**Fig. 5** Compressive stress induces the expression of *ENO2*, *HK2*, and *PFKFB3* genes in cancer-associated fibroblasts via c-Jun activation. **a** The dual-luciferase assay to measure the binding affinity of c-Jun to *ENO2*, *HK2*, and *PFKFB3* gene promoters. Potential c-Jun binding sites were predicted in each gene's 2 kb upstream promoter region using TRANSFAC database. The assay was performed in independent triplicates. **b** Chromatin immunoprecipitation assay to identify c-Jun binding to *ENO2*, *HK2*, and *PFKFB3* gene promoters in cancer-associated fibroblasts. The assay was independently performed six to nine times for each gene promoter. The compression-induced **c** mRNA expression ($n = 3$ independent experiments) and **d** protein expression of *ENO2*, *HK2*, and *PFKFB3* genes in the CAF cells exposed to compressive stress and/or treated with c-Jun inhibitor. **e** Compression-induced lactate production in the CAF cells exposed to compressive stress and/or treated with c-Jun inhibitor. For the inhibition of c-Jun phosphorylation, the CAF cells were pre-treated with 50 uM of SP600125 for 1 day ($n = 3$ independent experiments). The compression-induced **f** ENO2, HK2, and PFKFB3 protein expression and **g** lactate production ($n = 3$ independent experiments) in the c-Jun knockdown CAF cells. For c-Jun knockdown, the CAF cells were transfected with control or c-Jun shRNA. Compressive stress was given to the CAF cells after pretreatment or transfection. Error bars and *p*-values were determined by Whiskers (Min to Max) and unpaired two-tailed *t*-test, respectively. CS is the abbreviation of compressive stress. Source data are provided as Supplementary Data 1

the site 2 (S2, −578 to −571) of *HK2* gene promoter was not significantly increased by compression, but it was decreased with the pretreatment of SP600125. c-Jun binding to the site 1 (S1, −324 to −317) of *PFKFB3* gene promoter was significantly increased by compression compared to the control, whereas it was repressed with the pretreatment of SP600125 (Fig. 5b). The inhibition of c-Jun phosphorylation induced a significant down-regulation of ENO2, HK2, and PFKFB3 mRNA expression in compressed CAF cells (Fig. 5c). Similar to mRNA expression, the expression of ENO2, HK2, and PFKFB3 proteins was increased by compressive stress in CAF cells, which was reduced by the inhibition of c-Jun phosphorylation (Fig. 5d and Supplementary Fig. 3). Compression-induced promotion of lactate production was decreased by the inhibition of c-Jun phosphorylation (Fig. 5e). Further, c-Jun knockdown by shRNA decreased the compression-induced upregulation of ENO2, HK2, and PFKFB3 protein expression (Fig. 5f and Supplementary Fig. 4) and pro-motion of lactate production in CAF cells (Fig. 5g).

**Cancer progression was associated with the compression-induced upregulation of *ENO2*, *HK2*, or *PFKFB3* gene in cancer-associated fibroblasts**. The production of lactate, an end product of aerobic glycolysis[14], was increased in CAF cells by the compression-induced expression of *ENO2*, *HK2*, and *PFKFB3* genes. In addition, compressive stress was able to upregulate the expression of *SLC16A1* gene and induce lactate secretion from CAF cells to medium (Supplementary Fig. 5). Therefore, to investigate whether compression-induced lactate production in CAF cells contributes to cancer progression, the proliferation of breast cancer cells was analyzed after being treated with the conditioned medium (CM) from control CAF cells, compressed CAF cells, or compressed CAF cells treated with c-Jun inhibitor or transfected with shRNA against *ENO2*, *HK2*, or *PFKFB3* genes. In Fig. 6a, the proliferation of breast cancer cells, except for SK-BR-3 cells, was increased by the treatment of compressed CAF-derived CM, which was significantly decreased by c-Jun inhibition or the knockdown of *ENO2*, *HK2*, or *PFKFB3* genes in the compressed CAF cells. *PFKFB3* gene knockdown did not suppress the effect of compressed CAF-derived CM on BT-474 cell pro-liferation. Next, the expression of the epithelial to mesenchymal transition (EMT)- and angiogenesis-related genes, as potential factors for metastasis, in cancer cells were investigated in the same experimental settings. As shown in Fig. 6b, compressed CAF-derived CM treatment generally induced the upregulation of some of EMT inducers (*TWIST1*: Twist Family BHLH Tran-scription Factor 1, *SNAI1*: Snail Family Transcriptional Repressor 1, *ZEB1*: Zinc Finger E-Box Binding Homeobox 1, *ZEB2*: Zinc Finger E-Box Binding Homeobox 2), cadherins (*CDH1*: cad-herin1, and *CDH2*: cadherin2), matrix metallopeptidase (*MMP2*: Matrix Metallopeptidase 2), or angiogenesis-related factors (VEGFA: Vascular Endothelial Growth Factor A, B, C, or D) in breast cancer cells, which was significantly inhibited by the

knockdown of *ENO2*, *HK2*, or *PFKFB3* gene in CAF cells. Exceptionally, *SNAI1* gene was downregulated in BT-474 cells by compressed CAF-derived CM treatment, which was inversely upregulated by c-Jun inhibitor treatment to CAF cells or *ENO2*, *HK2*, or *PFKFB3* gene knockdown in CAF cells. The expression of *TWIST1*, *ZEB2*, *CDH2*, and *MMP2* genes were downregulated in SK-BR-3 cells by compressed CAF-derived CM treatment, which was upregulated by c-Jun inhibitor treatment to CAF cells or *ENO2* or *HK2* gene knockdown in CAF cells. Cadherin switching (*CDH1* downregulation and *CDH2* upregulation) was observed merely in MDA-MB-231 cells.

**The expression of *PFKFB3* gene was positively correlated with EMT- and angiogenesis-related gene expression and metastasis size in the breast cancer patient tissues with high compressive stress**. To investigate whether the findings in our in vitro model are observed in cancer patients, we had to classify breast cancer patient tissues into low- and high-compression groups. However, it was not possible to directly measure compressive stress from cancer patient tissues. According to previous studies, compressive stress is a major solid stress in tumor tissue and increases the epicenter and periphery of tumor[15]. Solid stress increases with tumor size[16] and its accumulation correlates with the expression of ECMs such as COL1A1, COL3A1, HAS2, and HAS3[17,18]. In the stiffened environment of tumor tissue, tumor growth results in compressive stress due to the resistance of hyaluronan[19]. Therefore, to establish a molecular basis to classify patient tissues into low- and high-compression groups, we analyzed the Pear-son's correlation between tumor size and ECM expression and found that tumor size is significantly and positively correlated with the expression of *COL1A1, HAS2*, and *HAS3* genes (Sup-plementary Fig. 6). Thus, breast patient tissues were classified into low-compression and high-compression group based on the expression of *HAS2*, *HAS3*, or *COL1A1* genes. Among the compression-upregulated metabolic genes, the expression of *ENO2* gene was significantly and positively correlated with *COL3A1* and *HAS1* genes, whereas *PFKFB3* genes were sig-nificantly and positively correlated with that of *COL1A1*, *HAS2*, and *HAS3* genes in breast cancer patient tissues (Fig. 7a). The expression of *ENO2*, *HK2*, and *PFKFB3* genes was significantly higher in the high-compression group than the low-compression one (Fig. 7b). The expression of EMT- or angiogenesis-related genes was significantly upregulated in the high-compression group compared to the low-compression one. The expression of *TWIST1*, *SNAI1*, *ZEB1*, *ZEB2*, *CDH1*, *CDH2*, and *MMP2* genes was higher in the high-compression group than the low-compression one (Fig. 7c). The expression of *VEGFA* and *VEGFB* genes was higher in the high-compression group than the low-compression one (Fig. 7d). The expression of *ENO2* gene was significantly and positively correlated with that of *CDH1* and *MMP2* genes. The expression of the *PFKFB3* gene was sig-nificantly and positively correlated with that of *TWIST1*, *SNAI1*,

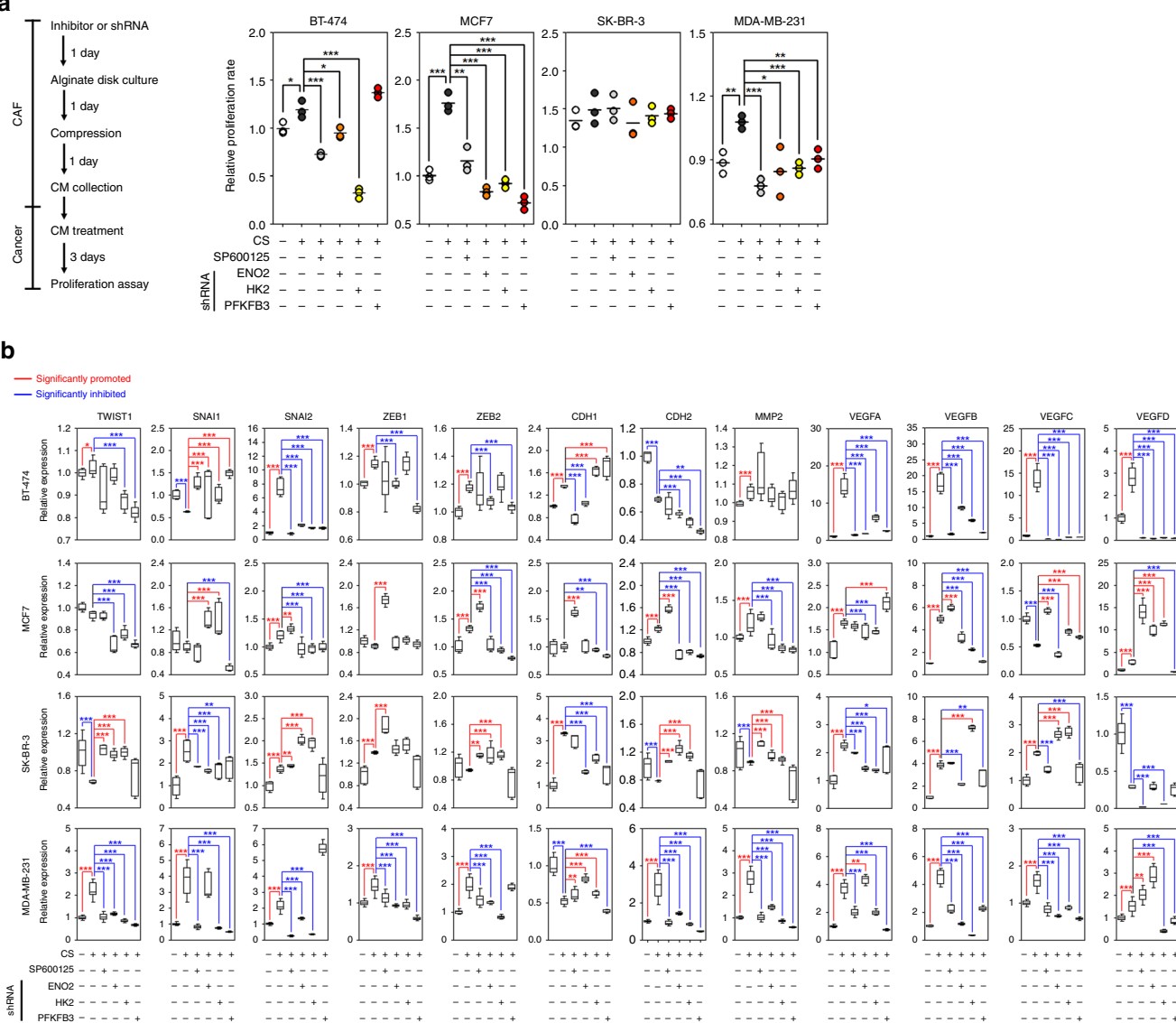

**Fig. 6** Cancer progression is associated with the compression-induced upregulation of *ENO2*, *HK2*, or *PFKFB3* gene in cancer-associated fibroblasts. **a** The proliferation assay (*n* = 3 independent experiments) and **b** the EMT-related gene expression analysis (*n* = 9 independent experiments) of the breast cancer cells treated with compressed CAF-derived CM. To block the compression-induced upregulation of *ENO2*, *HK2*, or *PFKFB3* gene, CAF cells were pre-treated with c-Jun inhibitor or transfected with the shRNAs against the corresponding genes before loading compression. Gene expression was analyzed using real-time PCR. Red and blue lines mean a significant promotion and inhibition, respectively. Error bars and *p*-values were determined by Whiskers (Min to Max) and unpaired two-tailed *t*-test, respectively. CS is the abbreviation of compressive stress. Source data are provided as Supplementary Data 1

*SNAI2*, *ZEB1*, *ZEB2*, *CDH1*, *CDH2*, and *MMP2* genes (Fig. 7e). The expression of *PFKFB3* gene was significantly and positively correlated with that of *VEGFA* and *VEGFB* genes (Fig. 7f). Invasion size and positive lymph node number were increased, but not significantly, in the high-compression group compared to the low-compression one (Fig. 7g, h). Metastasis size was significantly increased in the high-compression group compared to the low-compression one (Fig. 7i).

**PFKFB3 was positively correlated with compressive stress markers in a breast cancer clinical database.** Gene expression in solid tumor tissue is an outcome of both biochemical and mechanical signal transduction[20]. Therefore, the positive correlation of compression-upregulated metabolic genes with EMT-related and/or angiogenesis-related genes may be found in a clinical database. To confirm our assumption, we first confirmed the expression of the compression-induced metabolic genes

in breast cancer tissues using The Human Protein Atlas (version 18.1)[21] and then analyzed a METABRIC dataset in cBioportal[22,23]. As shown in Fig. 8a, among three compression-upregulated metabolic genes, the protein expression of *HK2* and *PFKFB3* genes were detected in breast cancer stromal tissues but not in normal tissue. HK2 protein expression was most strongly detected, whereas ENO2 protein expression barely observed in both normal and cancer tissues. To clarify the role of compression-induced upregulation of *ENO2*, *HK2*, or *PFKFB3* genes in breast cancer progression, we investigated genetic alteration of the genes. As shown in Fig. 8b, the alteration frequency of all the genes was below 3% of total breast cancer patients. Amplification was a major alteration in *ENO2* and *PFKFB3* genes, whereas missense mutations were observed in the *HK2* gene. Next, we analyzed the correlation of compression-upregulated metabolic genes with compressive stress markers (*COL1A1*, *COL3A1*, *HAS2*, and *HAS3* genes) and tumor size in

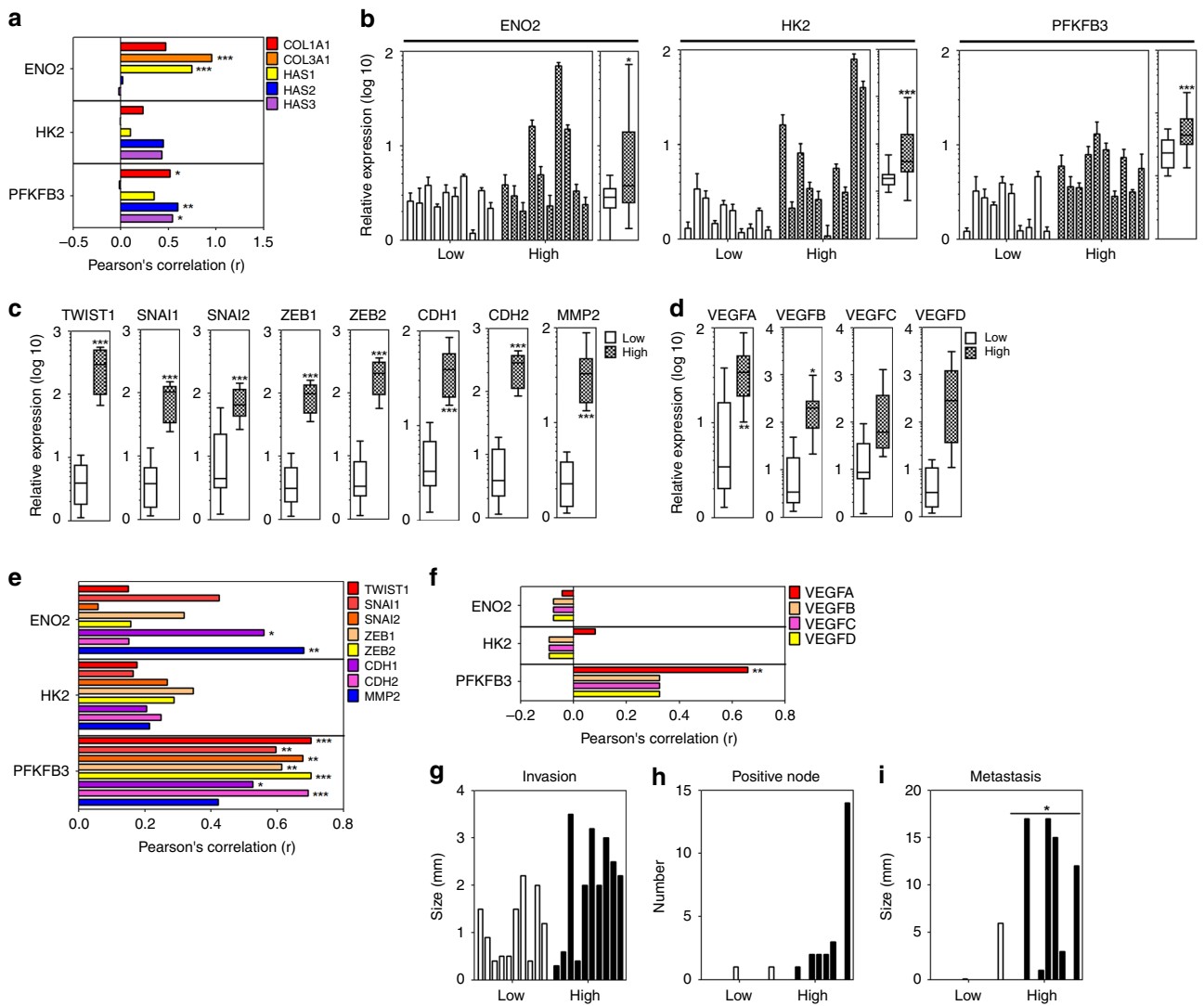

**Fig. 7** The expression of PFKFB3 is positively correlated with EMT- and angiogenesis-related gene expression and metastasis size in breast cancer patient tissues with high compressive stress. **a** Expressional correlation between the compression-upregulated metabolic genes and compressive stress markers. The comparative analysis of **b** the compression-upregulated metabolic genes, **c** EMT-related genes, and **d** angiogenesis-related genes between the low- and high-compression groups of breast cancer patient tissues. The expressional correlation of the compression-upregulated metabolic genes with **e** EMT-related genes or **f** angiogenesis-related genes. The comparative analysis of **g** invasion size, **h** lymph node number, and **i** metastasis size in breast cancer patient tissues. Breast cancer patient tissues were classified into the low- and high-compression groups based on the expression of compressive stress markers. Ten cases of patient tissues were analyzed for each group. The correlation coefficients and p-values were obtained by Pearson's correlation analysis. p-value for the relative expression of a gene was analyzed using unpaired two-tailed t-test. Source data are provided as Supplementary Data 1

breast cancer subtypes: luminal A (LumA), luminal B (LumB), Her2, and triple-negative (TN). In Fig. 8c, *PFKFB3* gene was significantly and positively correlated with *HAS2* gene in LumA. The *ENO2* gene was significantly and positively correlated with the *HAS3* gene, whereas *PFKFB3* gene was significantly and positively correlated with *COLA1*, *COL3A1*, and *HAS* genes in LumB. The *ENO2* gene was significantly and positively correlated with *COL1A1* gene, whereas the *PFKFB3* gene was significantly and positively correlated with *COL1A1*, *COL3A1*, *HAS2*, and *HAS3* in Her2. *HK2* gene was significantly and positively correlated with tumor size, whereas *PFKFB3* was significantly and positively correlated with *COL1A1*, *COL3A1*, and *HAS2* genes in TN.

**PFKFB3 was positively correlated with EMT and angiogenesis gene expression in a breast cancer clinical database**. Since *PFKFB3* gene is significantly and positively correlated with

compressive stress markers in all the subtypes of breast cancer, its correlation with EMT- or angiogenesis-related genes was further analyzed. In Fig. 9a, among the EMT-related genes, *SNAI2*, *ZEB1*, and *ZEB2* genes were significantly and positively correlated with *PFKFB3* gene in LumA. *SNAI2*, *ZEB1*, *MMP2* genes were significantly and positively correlated with *PFKFB3* gene in LumB. *SNAI2*, *ZEB1*, *ZEB2*, *CDH1*, *CDH2*, and *MMP2* genes were significantly and positively correlated with *PFKFB3* gene in Her2. *TWIST1*, *SNAI2*, *ZEB1*, *ZEB2*, and *MMP2* genes were significantly and positively correlated with *PFKFB3* gene in TN. In Fig. 9b, the expression of *ZEB2* gene was significantly upregulated in the PFKFB3-high LumA compared to the PFKFB3-low one. The expression of *MMP2* gene was significantly upregulated in the PFKFB3-high LumB compared to the PFKFB3-low one. The expression of *SNAI2*, *ZEB1*, *ZEB2*, *CDH2*, and *MMP2* genes was significantly upregulated in the PFKFB3-high Her2 compared to the

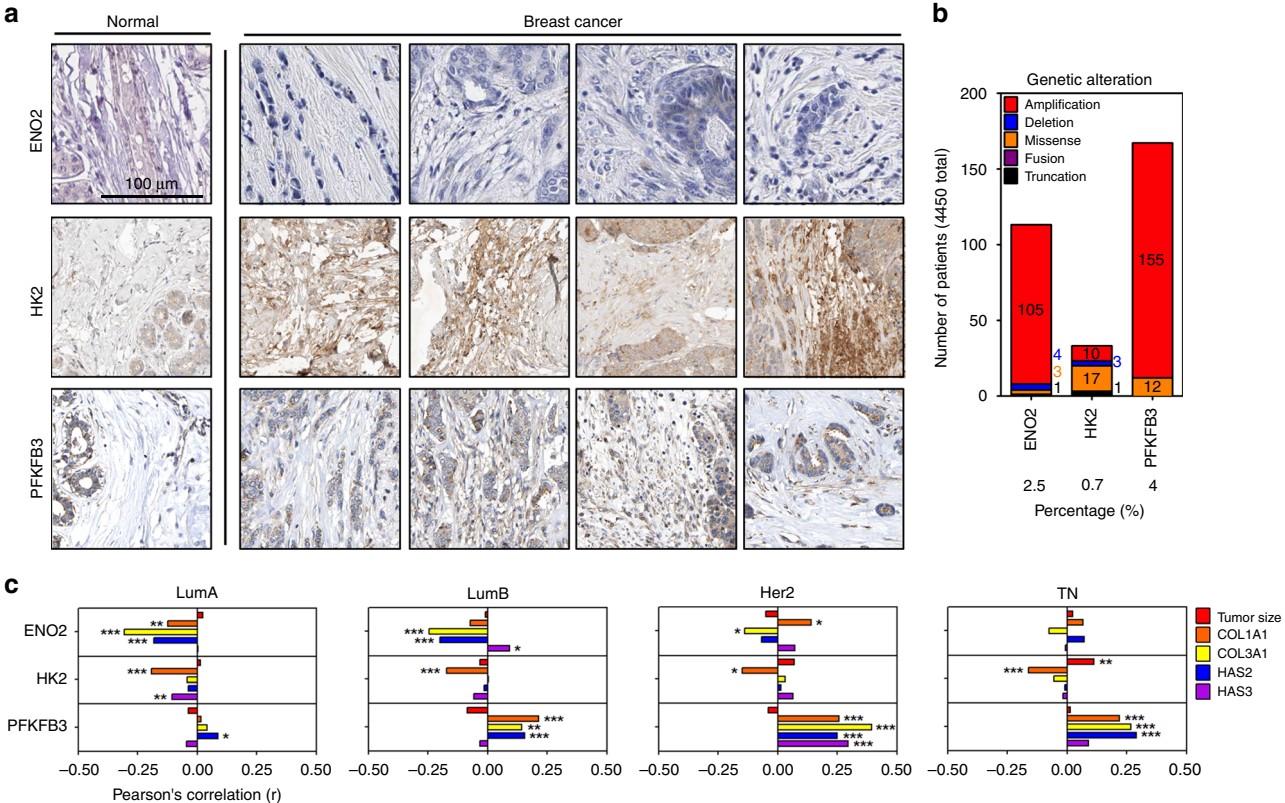

**Fig. 8** PFKFB3 expression is positively correlated with compressive stress marker expression in a breast cancer clinical database. **a** The expression of ENO2, HK2, and PFKFB3 proteins in breast cancer stroma. The images available from version 18.1. proteinatlas.org. ENO2 (normal: https://www.proteinatlas.org/ENSG00000111674-ENO2/tissue/breast#img, cancer: https://www.proteinatlas.org/ENSG00000111674-ENO2/pathology/tissue/breast+cancer#imid_202862), HK2 (normal: https://www.proteinatlas.org/ENSG00000159399-HK2/tissue/breast#img, cancer: https://www.proteinatlas.org/ENSG00000159399-HK2/pathology/tissue/breast+cancer#img), PFKFB3 (normal: https://www.proteinatlas.org/ENSG00000170525-PFKFB3/tissue/breast#img, cancer: https://www.proteinatlas.org/ENSG00000170525-PFKFB3/pathology/tissue/breast+cancer#img). **b** The genetic alteration analysis of *ENO2*, *HK2*, and *PFKFB3* genes in breast cancer patient tissues. **c** The expressional correlation of the compression-upregulated metabolic genes with compressive stress marker (including tumor size). The correlation coefficients and *p*-values were obtained by Pearson's correlation analysis (LumA = 699; LumB = 470; Her2 = 221; TN = 420). Source data are provided as Supplementary Data 1

PFKFB3-low one. The expression of *TWIST1*, *SNAI2*, *ZEB1*, *ZEB2*, and *MMP2* genes was significantly upregulated in the PFKFB3-high TN compared to the PFKFB3-low one. In Fig. 9c, among angiogenesis-related genes, *VEGFA* and *VEGFB* genes were significantly and positively correlated with *PFKFB3* gene in LumA. *VEGFB* gene was significantly and positively correlated with *PFKFB3* gene in LumB. *VEGFA* and *VEGFB* genes were significantly and positively correlated with *PFKFB3* gene in Her2. *VEGFA*, *VEGFB*, and *VEGFD* genes were significantly and positively correlated with *PFKFB3* gene in TN. In Fig. 9d, the expression of *VEGFB* and *VEGFD* genes was significantly upregulated in the PFKFB3-high LumA compared to the PFKFB3-low one. The expression of *VEGFB* gene was significantly upregulated in the PFKFB3-high LumB compared to the PFKFB3-low one. The expression of *VEGFA* gene was significantly upregulated in the PFKFB3-high Her2 compared to the PFKFB3-low one. The expression of *VEGFA*, *VEGFB*, and *VEGFD* gene was significantly upregulated in the PFKFB3-high TN compared to the PFKFB3-low one. Based on the correlation and expression analyses, PFKFB3 expression was associated with tumor progression in all breast subtypes. Therefore, we investigated whether PFKFB3 expression is related with a poor prognosis of breast cancer patients. In Fig. 9e, PFKFB3 expression is significantly associated with poor prognosis of breast cancer patients (low:1976 cases, high:1975 cases).

## Discussion

Tumor growth increases compressive stress in tissue, which is not only a prime phenomenon shared in solid tumors but also known to be associated with tumor progression. Therefore, for a better understanding of tumor progression mechanism, it seems necessary to study how compressive stress works in tumor progression. Unfortunately, however, most of the recent cancer research does not deal with compression-induced tumor progression. A few previous studies reported that compressive stress induces the invasive phenotypes of tumor cells[3] or inhibits the proliferation[24]. We also previously reported that compressive stress can induce the upregulation of tumor progression-related microRNAs[25] and epigenetically induce the expression of VEGFA, a proangiogenic factor, from CAF cells[6]. Nevertheless, very little is known about how compressive stress plays a role in tumor progression.

The activation of stromal glycolysis may be one of the major roles of compression-induced mechanotransduction in solid tumors. During tumor progression, the metabolic support of tumor stroma is known to be a critical process[26,27]. In solid cancer tissues, cancer cells are generally surrounded with a dense desmoplastic tissue consisting fibroblasts and extracellular matrix (ECM)[28]. Desmoplasia compresses vessels and thereby reduces oxygen and nutrient supply[3] as well as drug delivery[29]. In such a life-threatening environment, cancer cells may be evolutionarily

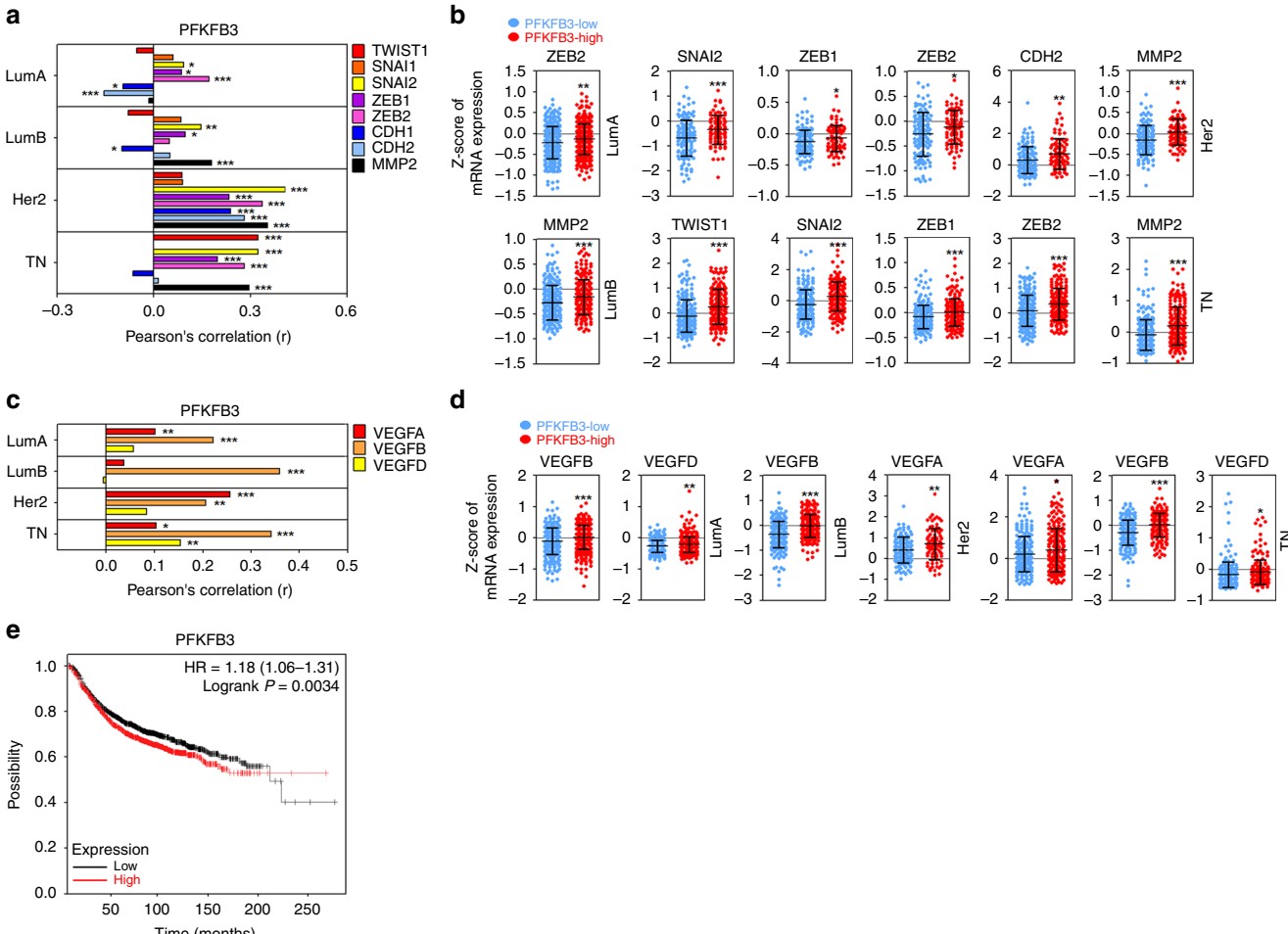

**Fig. 9** PFKFB3 expression is positively correlated with EMT and angiogenesis gene expression in a breast cancer clinical database. **a** The expressional correlation of PFKFB3 with EMT-related genes. **b** The comparative expression analysis of EMT-related genes between the low- and high-compression groups of breast cancer subtypes. **c** The expressional correlation of PFKFB3 with angiogenesis-related genes. **d** The comparative expression analysis of angiogenesis-related genes between the low- and high-compression groups of breast cancer subtypes. Gene expression and correlation were analyzed using a METABRIC dataset in cBioportal database. The correlation coefficients and p-values were obtained by Pearson's correlation analysis (LumA: PFKFB3-low = 324, PFKFB3-high = 375; LumB: PFKFB3-low = 242, PFKFB3-high = 228; Her2: PFKFB3-low = 135, PFKFB3-high = 86; TN: PFKFB3-low = 227, PFKFB3-high = 193). **e** The poor prognosis of breast cancer patients with PFKFB3 overexpression. The relation between PFKFB3 expression and breast cancer patient survival was analyzed using Kaplan–Meier Plotter (low:1976 cases, high:1975 cases)[53]. Source data are provided as Supplementary Data 1

able to survive through the compression-induced activation of stromal metabolism. Interestingly, mechanical compression is associated with the lactate production in cells[30]. In our study, a various degree of compressive stress was able to induce the expression of glycolysis-related metabolic genes (*ENO2*, *HK2*, and *PFKFB3*) and thereby increased lactate production in CAF cells. However, the expression of glycolysis-related genes was not dependent on the degree of compressive stress. A possible hypothesis for this phenomenon is that the responses to compressive stress is dynamic over time or different for each gene. For an example, the responses to transforming growth factor beta are diverse between cell types and environmental conditions[31]. The increase of lactate in tumor microenvironment contributes to EMT[32], metastasis[33], and angiogenesis[34,35] as well as tumor growth[36]. Similar with previous reports, the CM from compressed CAF cells contained a higher amount of lactate than that from control CAF cells and was able to induce the proliferation, EMT- and/or angiogenesis-related gene expression of some breast cancer cells, which were dependent on the expression of *ENO2*, *HK2*, and *PFKFB3* genes. In this study, we mainly dealt with the

compression-induced expression of *ENO2*, *HK2*, and *PFKFB3* genes in CAF cells. However, compressive stress is likely to promote the glycolysis of cancer cells. As shown in Fig. 4a, *HK2* and *PFKFB3* genes were found to be upregulated by compressive stress in BT-474 and SK-BR-3 cells, whereas many other glycolysis-related genes were upregulated in MDA-MB-231 cells. Considering that glycolysis is a prime mechanism for Warburg and reverse Warburg effect[37,38], compressive stress may play an important role in cancer metabolic rewiring. Based on our findings, the compression of tumor tissues during mammogram, a representative examination method for tracing and early recognition of breast cancer, could be considered as a potential risk factor to promote the reverse Warburg effect and tumor progression. In our previous study, the compression-induced alteration of gene expression returned to non-compression state in CAFs by decompression[6]. Therefore, compression during mammogram may temporarily promote the reverse Warburg effect although it is not likely to be a persistent effect.

The degree of compressive stress within tumor tissues may be exploited as an index for cancer diagnosis. Unfortunately,

however, it is not possible to directly measure solid stresses including compression from cancer patient tissues. Therefore, based on the expression of compressive stress markers (collagens and hyaluronan), we classified breast cancer patient tissues and a pre-existing clinical database into low- and high-compression group. In our study, the expression of the compression-upregulated metabolic genes, EMT-related genes, and angiogenesis-related genes was higher in the high-compression group of breast cancer patient tissues than low-compression one. In addition, metastasis size was significantly bigger in the high-compression group than low-compression one. Especially, of the compression-upregulated metabolic genes, *PFKFB3* gene was significantly and positively correlated with EMT- and angiogenesis-related genes as well as compressive stress markers. Since the outcomes from biochemical and mechanical signal transduction are both contained in solid tumor tissues[20], we further analyzed cancer database to investigate whether our findings are observed in a large pre-existing clinical samples. Among the compression-upregulated metabolic genes, the expression of *PFKFB3* gene was significantly and positively correlated with that of compressive stress markers (COL3A1, HAS2, and HAS3) in the LumB, Her2, and TN types of a clinical database (BRCA metabric). The expression of EMT- and angiogenesis-related factors was positively significantly correlated with that of PFKFB3 and higher in PFKFB3-high group than PFKFB3-low one. Breast cancer patients with high PFKFB3 expression showed a significant poorer prognosis than those with low PFKFB3 expression. Our analysis of the cancer database suggest the possibility that the high compressive stress state of patient tissues can be an index for cancer progression.

There is no suitable in vivo model to investigate compression-induced mechanotransduction for physiology and pathophysiology, which may be a major reason for the lack of research into compression-induced mechanotransduction in cancer research. In this study, we used alginate disk as a cell-encapsulating matrix. It is widely known that alginate allows the retention of extracellular matrix (ECM)[39], mimics many roles of ECM[40], and is easily depolymerized[41]. Particularly, the easy depolymerization of alginate disk enabled us to extract intact RNAs and proteins from the cells after loading compressive stress. The agarose matrix generally used in pre-existing in vitro compression models has to be treated with heat or chaotropic salts for depolymerization[42,43] and therefore cannot guarantee the integrity of RNAs and proteins[44,45]. RNA integrity is the most critical factor for the reliability of microarray-based profiling of transcriptome.

In our study, we showed that compression-induced mechanotransduction plays an important role in cancer metabolism using an in vitro compression model and the analysis of breast cancer patient tissues and a METABRIC dataset in cBioportal database. Since compression-induced mechanotransduction is a fundamental property shared in solid tumors, the compression-induced promotion of glycolysis has to be further investigated in various solid tumors. By doing that, we may be able to better understand about tumor progression and furthermore open a new era of cancer therapy.

## Methods

**Isolation of cancer-associated fibroblasts (CAFs) and cell cultures**. Human breast tumor tissues were acquired from four invasive ductal carcinoma (IDC) patients for the isolation of CAF cells. All patients donating the tissues had surgery at Severance Hospital of the Yonsei University Health System, South Korea. The research protocol was approved by the Severance Hospital Ethics Committee (IRB number 4-2008-0383). All participants signed consent forms and were informed of tissue use of comprehensive experiments on breast cancer. CAF cells were isolated as previously described[6,46]. Briefly, early stage invasive ductal carcinoma (IDC) tissues that were less than 10 mm in diameter were sliced and then digested overnight with a collagenase preparation (ISU ABXIS; Seoul, South Korea).

Digested tissue was filtered through a 70-μm cell strainer (SPL Life Science; Pocheon-si, South Korea). Cells were separated by Ficoll gradients, washed with phosphate-buffered saline (PBS), resuspended with Dulbecco's Modified Eagle's medium (DMEM)/F12 cell culture medium containing 20% (v/v) fetal bovine serum (FBS), 100 IU/mL penicillin, and 100 μg/mL streptomycin (Gibco BRL; Grand Island, NY), and cultured at 37 °C in a humidified incubator containing 5% $CO_2$. The fibrotic nature of the isolated cells was confirmed by microscopic determination of morphology and immunofluorescence characterization using antibodies against vimentin (Abcam; Cambridge, UK), alpha-smooth muscle actin (Santa Cruz Biotechnology, Dallas, TX), and cytokeratin (Dako; Glostrup, Denmark) (Supplementary Fig. 7). Breast cancer cell lines (BT-474, MCF7, SK-BR-3, MDA-MB-231) were purchased from the Korean Cell Line Bank (authenticated using morphology and STR profiling) and cultured with DMEM cell culture medium containing 10% (v/v) fetal bovine serum (FBS), 100 IU/ml penicillin, and 100 μg/ml streptomycin at 37 °C in a humidified incubator containing 5% $CO_2$. All cell lines used in this study were tested negative for mycoplasma contamination. Mycoplasma test was performed with MycoAlert® Mycoplasma Detection Kit (Lonza, Basel, Switzerland).

**Compression assay**. Compression assay was performed as previously described[6,25]. Briefly, as shown in Fig. 1b, to make the alginate disk containing cells, cell pellets ($5 \times 10^6$ cells, BT-474, MCF7, SK-BR-3, MDA-MB-231, and CAF cells) were resuspended with 500 ul of the 2× growth medium (DMEM or DMEM/F12) supplemented with 20% (v/v) fetal bovine serum (FBS), 200 IU/mL penicillin, and 200 μg/mL streptomycin (Gibco BRL; Grand Island, NY,), added with the same volume of 2% alginate solution, 20 mM of $CaCO_3$, and 50 mM of GDL, and then mixed gently using a 1 ml pipette. Since alginate polymerization was rapid in this condition, the mixture was immediately spread on the membrane (0.4 μm) of Transwell insert stand (6 well, SPL, Pocheon-si, Gyeonggi-do, South Korea), and incubated at 37 °C in a humidified incubator containing 5% $CO_2$ for 5 min. The alginate disk containing cells were washed with PBS twice, and then equilibrated with growth medium at 37 °C in a humidified $CO_2$ incubator for 1 h. Two milliliter of growth medium was added to Transwell insert stand (upper chamber) and lower chamber, respectively. Compression was performed by weight loading using empty or iron bead-filled cylinder (Fig. 1b-ⓐ and -ⓑ). In our experimental settings, the cylinders were held and guided by cylinder holder (Fig. 1b-ⓒ), which was critical for the accurate transfer of compressive stress to cells. The cell containing alginate disk was exposed to 0.386, 0.773, 1.546, 3.866, and 7.732 kPa at 37 °C in a humidified incubator containing 5% $CO_2$ for 24 h. 0.773 kPa (5.8 mmHg) is known to be the approximate compression value of a native breast tumor microenvironment[1,4]. The transfer of compressive stress to cells was indirectly confirmed by measuring the deformation of alginate disk (Fig. 1c). To collect the cells exposed to compressive stress, the alginate disk containing cells was depolymerized by gentle agitation with 40 mM EDTA for 3 min and then centrifuged at $250 \times g$ for 3 min. The resulting cell pellet was washed with PBS twice, centrifuged at $250 \times g$ for 3 min, and then used immediately in experiments, or stored at −80 °C.

**Transcriptomic alteration analysis**. *Functional annotation clustering analysis*: For functional annotation clustering, the genes commonly being above 2-fold upregulated or downregulated at all compressive stresses compared to the control (0 kPa) were first sorted from the microarray profiling data, and then analyzed using DAVID Bioinformatics Resources 6.7[7,47]. The functional annotation enrichment analysis was performed with the classification stringency of medium, high, and highest. The biological processes having over 1.3 of enrichment score (equivalent to a non-log scale 0.05) were considered to be valid.

*Supervised gene expression analysis*: For supervised gene expression analysis, the genes involved in the biological processes enriched from the functional annotation clustering were sorted from microarray profiling data based on the list of Amigo2 database[8,9]. Gene expression was presented using the dot distribution graph with relative expression values at 0.773 kPa (x-axis) and average relative expression values at all compressive stresses (y-axis). The genes with values greater than or equal to 2 both in x- and y-axes were counted in the upper-right corner of graphs, whereas the genes those less than −2 value in both in x- and y-axes were counted in the lower-left corner. For the tendency prediction, the compression-induced alteration of the BPs was compared between cells using the ratio of the number of upregulated genes divided by the number of downregulated genes. To compare gene expression between cell types, heatmap generation and hierarchical clustering were performed using MeV (version 4.9.0) with Pearson Correlation.

**Classification of patient tissues according to compressive stress**. Compressive stress is a major solid stress in tumor tissue and increases at both the epicenter and periphery of tumor[15]. Solid stress is positively correlated with the expression of ECMs such as HAS2, HAS3, COL1A1, and COL3A1[17,18]. Tumor growth results in compressive stress due to the resistance of hyaluronan[19]. Therefore, breast cancer patient tissues were relatively classified into low- and high-compression groups based on the z-score of collagen and hyaluronan expression.

**Microarray**. Total RNAs were extracted from uncompressed or compressed BT-474, MCF7, SK-BR-3, MDA-MB-231, and CAF cells using Trizol® (Invitrogen Life

Technologies; Grand Island, NY). RNA quality control (ratio of 28 s/18 s > 1.5, RIN > 7.0) was performed using an Agilent 2100 Bioanalyzer (Agilent Technologies; Santa Clara, CA). Total RNAs were reverse-transcribed and then analyzed on the SurePrint G3 Human Gene expression 8 × 60 K v2 Microarray (Agilent Technologies; Santa Clara, CA).

**Cell viability measurement**. The alginate disks containing cells were placed in each well of a 96-well plate, and then cultured at 37 °C in a humidified incubator containing 5% $CO_2$ for 1 day. Ten microliter of the cell count solution-8 (CCK-8, Dojindo Molecular Technologies, Inc, Rockville, MD) was added to each well of the plate. One hour after incubation, the absorbances at 450 and 650 nm were measured in triplicate using Spectramax plus 96/384 (MTX Lab Systems, Bradenton, FL). To calculate the relative cell viability, the absorbance values at 450 nm were subtracted with that at 650 nm, and then normalized to the absorbance of the control.

**Lactate assay**. The amount of lactate in cells was measured using Lactate assay kit (Sigma, St. Louis, MO) according to manufacturer's manual. Briefly, cells were lyzed with lactate assay buffer, and then centrifuged $13,000 × g$ for 10 min. 50 ul of supernatant was transferred to a well of a 96-well plate, and then 50 ul of the master reaction mix containing 46 ul of lactate assay buffer, 2 ul of lactate enzyme mix, and 2 ul of lactate probe was added to the well. Thirty minutes after incubation, absorbance was measured in triplicate at 570 nm using Spectramax plus 96/384 (MTX Lab Systems, Bradenton, FL). The amount of lactate in unknown samples was calculated from the standard curve made with known concentration of lactate and also compensated with the protein concentration of samples.

**Quantitative real-time PCR assays**. Quantitative PCR analysis was performed as previously reported[6]. Total RNAs were extracted with Trizol (Invitrogen; Carlsbad, CA) and then reverse-transcribed using the HyperScript™ Reverse Transcriptase (Geneall, Seoul, South Korea) in a PTC-200 Thermal Cycler (MJ Research, Reno, NV). The resulting cDNA (25 ng) was amplified using LaboPass™ SYBR Green Q Master (Cosmogenetech, Seoul, South Korea) for quantitative real-time PCR. PCR experiments were performed in triplicate. Primer sequences are provided as Supplementary Table 1. Real-time PCR was performed in a CFX Connect™ Real-Time PCR Detection System (Bio-Rad Laboratories; Hercules, CA). The expression of gene transcripts was normalized to the geomean values of endogenous glyceraldehyde-3-phosphate dehydrogenase (GAPDH), succineate dehydrogenase complex subunit A, flavoprotein (FP)(SDHA), and hypoxanthine phosphoribosyltransferase 1 (HPRT1)[48], and relative expression values were calculated according to the ΔΔCt method.

**Western blot**. As previously reported[6], cells were lysed in 50 µl of PRO-PREP Protein Extraction Solution (iNtRON Biotechnology; Seongnam-si, South Korea), homogenized using a 30-gauge needle, incubated for 30 min at 4 °C, and then centrifuged at $15,000 × g$ in a Centrige 5810 R (Eppendorf, Hamburg, Germany). After quantifying proteins in the extracts using the Bradford method, 20 µg protein was subjected to electrophoresis on 10% polyacrylamide gels in Tris/glycine (Invitrogen®, Carlsbad, CA), transferred to a PVDF membrane (Millipore Corporation, Billerica, MA), and then probed with primary antibodies against ENO2, HK2, PFKFB3 (Abcam; Cambridge, UK, 1:000), JNK, p-JNK, c-Jun, p-c-Jun, and beta-Actin (Santa Cruz Biotechnology, Dallas, TX, 1:000). Primary antibodies were detected by horseradish peroxidase (HRP)-conjugated secondary antibodies (Invitrogen®, Carlsbad, CA, 1:10000) and visualized using enhanced chemiluminescence reagents (Santa Cruz Biotechnology, Dallas, TX). The intensity ratio (IR) of western blot bands was measured using Image J 1.50i[49].

**Firefly luciferase reporter constructs and luciferase assays**. The 2 kb upstream sequences of human ENO2, HK2, and PFKFB3 genes were amplified from the genomic DNA of CAF cells. To make deletion mutant constructs, the putative binding sites for c-Jun were deleted in the promoter sequence of each gene by an overlap extension PCR method[50]. The wild and deletion mutant types of gene constructs were cloned into pGL3 vector. The integrity and orientation of the inserts were confirmed by sequencing. For luciferase assay, wild or deletion mutant plasmid was co-transfected with pRL-TK into HEK293T ($2 × 10^5$) cells in a 6-well plate using Lipofectamine® LTX with Plus™ Reagent (Invitrogen®, Carlsbad, CA). The cells were harvested 48 h after transfection, and luciferase activity was measured using a dual-luciferase reporter assay system (Promega; Madison, WI).

**Chromatin immunoprecipitation assay**. Chromatin immunoprecipitation (Chip) assay was performed with EpiQuick™ Chromatin immunoprecipitation kit (Epigentek, Farmingdale, NY)) according to the manufacturer's manual as we previously reported[6]. Briefly, the alginate disks containing CAF cells were exposed to 0.773 kPa for 24 h and then depolymerized with 102 mM EDTA. After being washed twice with PBS using centrifugation, the CAF cells were resuspended with the fresh culture medium containing 1% formaldehyde (final concentration) and incubated at room temperature (RT) for 10 min on a rocking platform

(50–100 rpm) for fixation. The fixed CAF cells were washed three times with ice-cold PBS, lyzed with PRO-PREP Protein Extraction Solution (iNtRON Biotechnology; Seongnam-si, South Korea), and then centrifuged to collect supernatant. The supernatant was sonicated for DNA shearing, transferred to the well coated with anti-c-Jun antibody (Santa Cruz Biotechnology, Dallas, TX), incubated at RT for 1 h. After washing PBS, proteinas K was treated and incubated at 65 °C for 90 min for DNA elution. DNA was collected using spin column, and analyzed by real-time PCR. The relative binding of c-Jun to each gene promoter was calculated using the ΔΔCt method. The results were normalized to those of input DNA[51].

**Gene knockdown**. ENO2 shRNA (MISSION® TRC shRNA TRCN0000157686), HK2 shRNA (MISSION® TRC shRNA TRCN0000037672), and PFKFB3 shRNA (MISSION® TRC shRNA TRCN0000007338) were transfected to CAF cells using Lipofectamine® LTX with Plus™ Reagent (Invitrogen®, Carlsbad, CA).

**Statistics and reproducibility**. Statistical significance was determined using t-test (two-tailed) and Pearson's correlation coefficient. Results were considered to be significant at $p < 0.05$. All statistical analyses were performed using Prism 6 for Windows (GraphPad Software, Inc.; La Jolla, CA). Data were presented as the mean ± standard deviation. Asterisks were used to indicate p-values: one for $p < 0.05$, two for $p < 0.01$, and three for $p < 0.001$.

**Reporting summary**. Further information on research design is available in the Nature Research Reporting Summary linked to this article.

## Data availability

All data supporting the findings of this study are available in Supplementary Information and Supplementary Data 1 and 2. Microarray raw data generated in study have been deposited into the GEO database (http://www.ncbi.nlm.nih.gov/geo/) under accession number GSE133134. Previously generated data analysed here are available in The Human Protein Atlas (version 18.1)[21] and in METABRIC dataset in cBioportal[22,23]

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

## Acknowledgements

This study was supported by the Mid-Career Researcher Program (No.2019R1-A2B5B01069934; NHC) and Basic Research in Science and Engineering (No.2016R1-D1A1B03932310; BGK) through a National Research Foundation of Korea grant

## Author contributions

All authors of this paper have read and approved the final version submitted, and have directly participated in the planning, execution, or analysis of the study. B.G.K. designed the experiments, mainly performed the experiments, analyzed the data, and drafted and wrote the manuscript. J.S.S. performed western blot analysis and helped compression assays. Y.J. performed real-time PCR analysis and helped culturing cancer cells. Y.J.C. measured the compressive stress value of breast cancer tissues and collected them. S.K. performed dual-luciferase assays. H.H.H. and J.H.L. helped to isolate primary fibroblasts and prepared them for assays. N.H.C. designed the experiments, interpreted the data, and prepared the manuscript by writing and organizing the figures.

## Additional information

**Competing interests:** The authors declare no competing interests.

