## [Peer Review File · Communications Biology]

Reviewers' comments:

Reviewer #1 (Remarks to the Author):

This is an excellent article that deserves to be published in *Communications Biology*, with only minor modifications.

My suggestion is that the authors should cite the following relevant paper: Lisanti et al., JNK1 stress signaling is hyper-activated in high breast density and the tumor stroma: connecting fibrosis, inflammation, and stemness for cancer prevention. *Cell Cycle*. 2014;13(4):580-99.

Based on their findings, they should also discuss the intriguing possibility that the mammogram test could induce the reverse Warburg effect and actually promote the onset of high breast density and breast cancer itself, due to the stress of compression.

Reviewer #2 (Remarks to the Author):

The authors study compressive stresses in tumors and their consequences in the metabolism of cancer cells and cancer associated fibroblasts, an important and emerging topic in tumor microenvironment. The authors have improved the in vitro model of compressive stresses, by using alginate gel instead of conventional method that uses agarose, which is helpful in cell isolation, and consequent protein/mRNA analyses. This model system is useful to study the consequences of compressive stresses, however, it has to be better characterized. The authors also rely heavily on gene expression rather than protein level analysis. The ms will be improved significantly if the major changes of expression can be reported at the protein level as well. Other major and minor comments with no particular order:

The stress values are reported in RCU, an unusual unit. Please change all the stress values to the unit of Pa. The authors cited ref 1 for the pathophysiological range of stress value, a non-relevant paper in this aspect. The authors use compressive stresses that seem to be too high. According to *Nat. Biomed. Eng.* 2017 by Jain group, pathophysiological range of stress value in breast tumors is between 0 and 1 kPa (7.5 mmHg) including the tensile stresses. The upper limit of compressive stresses are even lower.

The description and characterization of compression assay is not detailed and complete. The pore sizes of the membrane is not mentioned. As shown in Fig. 1C, the gel thickness changes by a factor of 10 after compression. This large change of density results in changes in pore size, and consequently changes in diffusion of media and oxygen in the gel, which can affect the cell metabolisms. Because oxygen or nutrient limitations would dramatically affect the analyses, the authors should demonstrate that the cells are not experiencing such limitations due to the presence of the compression weight and confinement.

It would be interesting to know why different cell lines and primary CAFs show such a large heterogeneous respond to compressive stresses as shown in Fig. 2 and 3. The authors need to mention how repeatable their results are with the same cell type.

The ms lacks mechanistic studies. c-jun is claimed to regulate response of genes such as ENO2, HK2, and PFKFB3. However, the hypothesis is tested via only correlations in gene expressions, and intervention via c-Jun inhibitor SP600125. The authors need to test their hypothesis with more specific

interventions, e.g., KO and KD interventions. The pharmacologic inhibition of c-jun results in lower JNK, c-jun, ENO2, HK2, PFKFB3 (Fig. 5D). But in some of these genes, they still show a dose-dependent response to compressive stress, and some don't. It would be useful to show the quantification of the western blot for a clearer results, and discuss the difference in dose-dependent response.

The human data are not very informative or supportive of the authors' hypothesis because it is not possible to know what levels of stress were present in the samples. The authors misunderstood the difference between stiffness (measured via SWE) and compressive stresses when they present the clinical data. They have classified patients based on shear wave elastography (SWE), a method that measures stiffness (rigidity) of the tissue. They, however, claim they have classified based on the high and low compressive stresses, which is not correct.

In Fig 7, it is not possible to know how many CAFs and how many cancer cells contributed to the signal; therefore, the results are difficult to interpret in light of the rest of the study.

In Fig 7, it is not possible to know how many CAFs and how many cancer cells contributed to the signal; therefore, the results are difficult to interpret in light of the rest of the study.

More technical comments:

the authors need to present the western blots for shRNA KD to show the quality of KD.

The authors also need to have more controlled experiments: the group with no compression but treated with SP600125 is missing in Fig. 5E.

The frequent use of broken y axes is overused, confusing and unnecessary in most cases.

COMMSBIO-19-0007A

Thank you for your valuable suggestions.

According to reviewers' comments, we carefully revised our initial manuscript.

The changes we made in response to reviewer's comments are as follows.

To Reviewer #1:

Thank you for your review and comments.

Comment 1.

This is an excellent article that deserves to be published in Communications Biology, with only minor modifications. My suggestion is that the authors should cite the following relevant paper: Lisanti et al., JNK1 stress signaling is hyper-activated in high breast density and the tumor stroma: connecting fibrosis, inflammation, and stemness for cancer prevention. Cell Cycle. 2014;13(4):580-99. Based on their findings, they should also discuss the intriguing possibility that the mammogram test could induce the reverse Warburg effect and actually promote the onset of high breast density and breast cancer itself, due to the stress of compression.

Comment 1-1.

My suggestion is that the authors should cite the following relevant paper: Lisanti et al., JNK1 stress signaling is hyper-activated in high breast density and the tumor stroma: connecting fibrosis, inflammation, and stemness for cancer prevention. Cell Cycle. 2014;13(4):580-99.

Response:

Since the reviewer's comment was fairly reasonable, we cited the article of Lisanti et al. in our manuscript as follows.

Page 7, line 171

According to a previous study, c-Jun and c-Fos is known to be activated by mechanical stress¹⁰ and fibrosis is associated with JNK (c-Jun terminal kinase)-induced stress signaling¹¹. It is therefore plausible that compressive stress promotes the expression of ENO2, HK2, and PFKFB3 genes via the

activation of c-Jun and/or c-Fos.

Comment 1-2.

Based on their findings, they should also discuss the intriguing possibility that the mammogram test could induce the reverse Warburg effect and actually promote the onset of high breast density and breast cancer itself, due to the stress of compression.

Response:

As agreed with the reviewer's opinion, we discussed the possibility in the discussion section.

Page 15, line 357 - 362

Based on our findings, the compression of tumor tissues during mammogram, a representative examination method for tracing and early recognition of breast cancer, could be considered as a potential risk factor to promote the reverse Warburg effect and tumor progression. In our previous study, the compression-induced alteration of gene expression returned to non-compression state in CAFs by decompression⁶. Therefore, compression during mammogram may be likely to promote the reverse Warburg effect temporarily but not consistently.

To Reviewer #2:

Thank you for your review and comments.

Comment 1.

The stress values are reported in RCU, an unusual unit. Please change all the stress values to the unit of Pa. The authors cited ref 1 for the pathophysiological range of stress value, a non-relevant paper in this aspect. The authors use compressive stresses that seem to be too high. According to Nat. Biomed. Eng. 2017 by Jain group, pathophysiological range of stress value in breast tumors is between 0 and 1 kPa (7.5 mmHg) including the tensile stresses. The upper limit of compressive stresses are even lower.

Comment 1-1

The stress values are reported in RCU, an unusual unit. Please change all the stress values to the unit of Pa.

Response:

Values in Relative Compression Unit (RCU) were all replaced into those in kPa.

1. Figures and Tables

1-1. Figure 1C and 1D

1-2. Figure 3A

1-3. Figure 4A, 4B, and 4D

1-4. Figure 5D

1-5. Supplementary Figure S2A (previously S1A).

2. Legends of Figures and Tables (including Supplementary Figures and Tables)

Page 25, line 604

Figure 1. An in vitro compression model using alginate disk.

Before

E. The immunofluorescence image of the alginate disk containing cells. The CAF cells labeled with CellTracker Red were embedded into alginate disk, placed on a well of 6-well tissue culture plate, and incubated in growth medium for 1h, and exposed to 5 RCU for 1day.

After

E. The immunofluorescence image of the alginate disk containing cells. The CAF cells labeled with CellTracker Red were embedded into alginate disk, placed on a well of 6-well tissue culture plate, and incubated in growth medium for 1h, and exposed to 3.866 kPa for 1day.

Page 25, line 607

Before

Figure 2. The functional annotation clustering of compression-induced transcriptomic alteration.

The genes commonly being over 2-fold upregulated or downregulated at all RCUs were analyzed by using DAVID. The presented BPs had over 1.3 enrichment score and were found in at least 3 types of cells. For comparison, all the BPs were displayed together using Circos ³⁸.

After

Figure 2. The functional annotation clustering of compression-induced transcriptomic alteration.

The genes commonly being over 2-fold upregulated or downregulated at all compressive stresses were analyzed by using DAVID. The presented BPs had over 1.3 enrichment score and were found in at least 3 types of cells. For comparison, all the BPs were displayed together using Circos ³⁸.

Page 26, line 616 - 617

Before

Figure 3. The compression-induced expression alteration of the enriched biological process-related genes.

The x- and y-axis represent the average values of gene expression at all RCUs and the values of gene expression at 1 RCU, respectively.

After

Figure 3. The compression-induced expression alteration of the enriched biological process-related genes.

The x- and y-axis represent the average values of gene expression at all compressive stresses and the

values of gene expression at 1 compressive stress, respectively.

Page 26, line 630 - 636

Before

Figure 4. The compression-induced promotion of glycolysis in cancer-associated fibroblasts was associated with the upregulation of ENO2, HK2, and PFKFB3 genes.

C) The compression-induced upregulation of ENO2, HK2, and PFKFB3 mRNAs. Total RNA was extracted from the CAF cells exposed to 1 RCU for 1 day, reverse-transcribed, and then analyzed using real-time PCR. D) The compression-induced promotion of lactate production. CAF cells were exposed to 0, 0.5, 1, 2, 5, and 10 RCU for 1 day. E) The decreased production of lactate by gene knockdown. CAF cells were transfected with the shRNAs against ENO2, HK2, or PFKFB3 gene. F) The effect of gene knockdown on the compression-induced promotion of lactate production. CAF cells were transfected with the shRNAs against ENO2, HK2, or PFKFB3 gene, and then exposed to 1 RCU. CS is the abbreviation of compressive stress.

After

Figure 4. The compression-induced promotion of glycolysis in cancer-associated fibroblasts was associated with the upregulation of ENO2, HK2, and PFKFB3 genes.

C) The compression-induced upregulation of ENO2, HK2, and PFKFB3 mRNAs. Total RNA was extracted from the CAF cells exposed to 0.773 kPa for 1 day, reverse-transcribed, and then analyzed using real-time PCR. D) The compression-induced promotion of lactate production. CAF cells were exposed to 0.386, 0.773, 1.546, 3.866, and 7.732 kPa for 1 day. E) The decreased production of lactate by gene knockdown. CAF cells were transfected with the shRNAs against ENO2, HK2, or PFKFB3 gene. F) The effect of gene knockdown on the compression-induced promotion of lactate production. CAF cells were transfected with the shRNAs against ENO2, HK2, or PFKFB3 gene, and then

exposed to 0.773 kPa. CS is the abbreviation of compressive stress.

Page 27, line 649

Before

Figure 5. Compressive stress induced the expression of ENO2, HK2, and PFKFB3 genes in cancer-associated fibroblasts via c-Jun activation.

E) Compression-induced lactate production in the CAF cells exposed to compressive stress and/or treated with c-Jun inhibitor. For the inhibition of c-Jun phosphorylation, the CAF cells were pre-treated with 50 μ M of SP600125 for 1 day and then exposed to 1RCU for 1 day.

After

Figure 5. Compressive stress induced the expression of ENO2, HK2, and PFKFB3 genes in cancer-associated fibroblasts via c-Jun activation.

E) Compression-induced lactate production in the CAF cells exposed to compressive stress and/or treated with c-Jun inhibitor. For the inhibition of c-Jun phosphorylation, the CAF cells were pre-treated with 50 μ M of SP600125 for 1 day and then exposed to 0.773 kPa for 1 day.

Page 29, line 700

Before

Supplementary Figure S2. Compression-induced lactate secretion from CAF cells.

A) The compression-induced upregulation of SLC16A1 gene in CAF cells. B) Lactate assay with the CM from CAF cells with or without compression. For compression, CAF cells were exposed to 1RCU for 1 day.

After

Supplementary Figure S2. Compression-induced lactate secretion from CAF cells.

A) The compression-induced upregulation of SLC16A1 gene in CAF cells. B) Lactate assay with the CM from CAF cells with or without compression. For compression, CAF cells were exposed to 0.773 kPa for 1 day.

3. Results

Page 4, line 85 - 91

Before

In Figure 1C, alginate disk thickness was proportionally decreased to the degree of compressive stress (0 to 5 RCU, 1RCU is 5.8 mmHg - the compressive stress value of a native tumor microenvironment^{1, 4}). However, 10 RCU of compressive stress frequently broke alginate disks. Next, it was examined whether cell viability is associated with the degree of compressive stress. As shown in Figure 1D, cell viability was not significantly different from 0 to 5 RCUs, but it was significantly decreased at 10 RCU. Alginate disk was suitable to contain cells under compressive conditions. In Figure 1E, most cells (RFP-positive) were within an alginate disk under compression (5 RCU for 1 day).

After

In Figure 1C, alginate disk thickness was proportionally decreased to the degree of compressive stress (0 to 7.732 kPa, 0.773 kPa - the compressive stress value of a native tumor microenvironment⁴). However, 7.732 kPa of compressive stress frequently broke alginate disks. Next, it was examined whether cell viability is associated with the degree of compressive stress. As shown in Figure 1D, cell

viability was not significantly different from 0 to 3.866 kPa, but it was significantly decreased at 7.732 kPa. Alginate disk was suitable to contain cells under compressive conditions. In Figure 1E, most cells (RFP-positive) were within an alginate disk under compression (3.866 kPa for 1 day).

Page 5, line 100 - 101

Before

To investigate whether compressive stress induces the biological processes (BPs) possibly related to tumor progression, the genes commonly being over 2-fold upregulated or downregulated at all RCUs (0.5, 1, 2, 5, and 10) were sorted from the transcriptome profiling data of breast cancer cell lines (BT-474: luminal B, MCF7: luminal A, SK-BR-3: HER2, and MDA-MB-231: triple negative) and 4 patient-derived CAF cells (from invasive ductal carcinoma, stage 1) and then analyzed it using The Database for Annotation, Visualization, and Integrated Discovery (DAVID) Bioinformatics Resources 6.7⁷.

After

To investigate whether compressive stress induces the biological processes (BPs) possibly related to tumor progression, the genes commonly being over 2-fold upregulated or downregulated at all compressive conditions (0.386, 0.773, 1.546, 3.866, and 7.732 kPa) were sorted from the transcriptome profiling data of breast cancer cell lines (BT-474: luminal B, MCF7: luminal A, SK-BR-3: HER2, and MDA-MB-231: triple negative) and 4 patient-derived CAF cells (from invasive ductal carcinoma, stage 1) and then analyzed it using The Database for Annotation, Visualization, and Integrated Discovery (DAVID) Bioinformatics Resources 6.7⁷.

Page 5, line 120

Before

For the supervised validation of the functional enrichment clustering, based on the gene ontology from Amigo 2^{8,9}, we sorted the genes involved in the enriched BPs from the transcriptome profiling data and then analyzed their expression using dot distribution graphs to present both gene expression values at 1 RCU and average gene expression values at all RCUs.

After

For the supervised validation of the functional enrichment clustering, based on the gene ontology from Amigo 2^{8,9}, we sorted the genes involved in the enriched BPs from the transcriptome profiling data and then analyzed their expression using dot distribution graphs to present both gene expression values at 0.773 kPa and average gene expression values at all compressive conditions.

Page 6, line 144

Before

In Figure 4A, *ENO2* (enolase 2), *HK2* (hexokinase 2), and *PFKFB3* (6-Phosphofructo-2-Kinase/Fructose-2,6-Biphosphatase 3) genes were commonly upregulated at all RCUs in 4 patient-derived CAF cells.

After

In Figure 4A, *ENO2* (enolase 2), *HK2* (hexokinase 2), and *PFKFB3* (6-Phosphofructo-2-Kinase/Fructose-2,6-Biphosphatase 3) genes were commonly upregulated at all compressive conditions in 4 patient-derived CAF cells.

Page 7, line 154

Before

The compression-induced upregulation of *ENO2*, *HK2*, and *PFKFB3* genes was confirmed in the CAF cells exposed to 1 RCU, the compressive stress value of a native tumor microenvironment, using real-time PCR analysis.

After

The compression-induced upregulation of *ENO2*, *HK2*, and *PFKFB3* genes was confirmed in the CAF cells exposed to 0.386 kPa, the compressive stress value of a native tumor microenvironment, using real-time PCR analysis.

Page 7, line 159 - 160

Before

As shown in Figure 4D, the amount of lactate was significantly increased at 0.5, 1, 2, 5, and 10 RCUs compared to the control (the CAF cells at 0 RCU).

After

As shown in Figure 4D, the amount of lactate was significantly increased at 0.386, 0.773, 1.546, 3.866, and 7.732 kPa compared to the control (the CAF cells at 0 kPa).

4. Materials and Methods

Page 17, line 417 -420

Before

The cell containing alginate disk was exposed to 0.5, 1, 2, 5, or 10 relative compression unit (RCU) at 37 °C in a humidified incubator containing 5% CO₂ for 24 h. One RCU was defined to be 5.8 mmHg (~0.773 kPa)^{1, 4}, which is the approximate compression value of a native breast tumor microenvironment.

After

The cell containing alginate disk was exposed to 0.386, 0.773, 1.546, 3.866, and 7.732 kPa at 37 °C in a humidified incubator containing 5% CO₂ for 24 h. 0.773 kPa (5.8 mmHg) is known to be the approximate compression value of a native breast tumor microenvironment^{1,4}.

Page 17, line 428 - 429

Before

A. Functional annotation clustering analysis. For functional annotation clustering, the genes commonly being above 2-fold upregulated or downregulated at all RCUs compared to the control (0 RCU) were first sorted from the microarray profiling data, and then analyzed using DAVID Bioinformatics Resources 6.7^{7,37}.

After

A. Functional annotation clustering analysis. For functional annotation clustering, the genes commonly being above 2-fold upregulated or downregulated at all compressive stresses compared to

the control (0 kPa) were first sorted from the microarray profiling data, and then analyzed using DAVID Bioinformatics Resources 6.7^{7,37}.

Page 18, Line 436 - 437

Before

Gene expression was presented using the dot distribution graph with relative expression values at 1 RCU (x-axis) and average relative expression values at all RCUs (y-axis).

After

Gene expression was presented using the dot distribution graph with relative expression values at 1 compressive stress (x-axis) and average relative expression values at all compressive stresses (y-axis).

Comment 1-2.

The authors cited ref 1 for the pathophysiological range of stress value, a non-relevant paper in this aspect.

Response:

As the reviewer's comment was fairly reasonable, we removed the citation.

Comment 1-3.

The authors use compressive stresses that seem to be too high. According to Nat. Biomed. Eng. 2017

by Jain group, pathophysiological range of stress value in breast tumors is between 0 and 1 kPa (7.5 mmHg) including the tensile stresses. The upper limit of compressive stresses are even lower.

Response:

According to previous reports, Compressive growth-induced stress in the interior of murine tumors ranged from 2.8 to 60.1 mmHg (**0.37–8.01 kPa**, Proc Natl Acad Sci U S A. 2012 Sep 18; 109(38): 15101–15108). Compressive stress in multicellular tumor spheroids in vitro fall in the range of 28–120 mmHg (**3.7–16.0 kPa**) (Nat Biotechnol. 1997 Aug;15(8):778-83, PLoS One. 2009;4(2):e4632). Based on the reports, the range of compressive stress (0 to 7.732 kPa) we used in our study may be acceptable. After the unsupervised and supervised analyses of compression-induced gene expression, we mainly used a compressive stress of 0.773 kPa for most experiments.

Comment 2.

The description and characterization of compression assay is not detailed and complete. The pore sizes of the membrane is not mentioned. As shown in Fig. 1C, the gel thickness changes by a factor of 10 after compression. This large change of density results in changes in pore size, and consequently changes in diffusion of media and oxygen in the gel, which can affect the cell metabolisms. Because oxygen or nutrient limitations would dramatically affect the analyses, the authors should demonstrate that the cells are not experiencing such limitations due to the presence of the compression weight and confinement.

Comment 2-1.

The description and characterization of compression assay is not detailed and complete.

Response:

Page 16, line 403 - page 17, line 424

The detailed description of compression assay is in the Materials and Methods.

Comment 2-2.

The pore sizes of the membrane is not mentioned.

Response:

The pore size of the membrane was written in Figure 1B and the section of compression assay in Materials and Methods. **(page 17, line 410)**

Comment 2-3.

the authors should demonstrate that the cells are not experiencing such limitations due to the presence of the compression weight and confinement.

Response:

The result was provided as Supplementary Figure S1 and described in the result section as follows.

Supplementary Figure S1.

Page 4, line 91 - 96

Alginate disk deformation by compressive stress may cause the limitation of oxygen and nutrients which affects gene expression in cells. Therefore, the diffusion rate of Ponceau S (a red-colored dye having a molecular weight of 750 Da), which is a bigger molecule than all medium components (less than 500 Da), was measured from the alginate disks exposed to different degrees of compressive stress. In Supplementary Figures S1, the degree of compressive stress tested did not affect the diffusion rate of Ponceau S on the alginate disks.

Comment 3.

It would be interesting to know why different cell lines and primary CAFs show such a large heterogeneous response to compressive stresses as shown in Fig. 2 and 3. The authors need to mention how repeatable their results are with the same cell type.

Response:

It may be natural that different types of cells show heterogeneous responses to compressive stress. Transforming growth factor beta (TGF- β) can be a representative example of this phenomenon. The physiological responses to TGF- β stimulation are diverse among different cell types and environmental conditions (FEBS Lett. 2012 Jul 4; 586(14): 1921–1928.). To find a general response to compressive stress, we analyzed the commonly up- or down-regulated genes at all five different compressive stress conditions. Therefore, the patterns of the gene expression analyzed in our study were reproducible in five separated experiments.

Comment 4.

The ms lacks mechanistic studies. c-jun is claimed to regulate response of genes such as ENO2, HK2, and PFKFB3. However, the hypothesis is tested via only correlations in gene expressions, and intervention via c-Jun inhibitor SP600125. The authors need to test their hypothesis with more specific interventions, e.g., KO and KD interventions. The pharmacologic inhibition of c-jun results in lower JNK, c-jun, ENO2, HK2, PFKFB3 (Fig. 5D). But in some of these genes, they still show a dose-dependent response to compressive stress, and some don't. It would be useful to show the quantification of the western blot for a clearer results, and discuss the difference in dose-dependent response.

Comment 4-1.

The ms lacks mechanistic studies. c-jun is claimed to regulate response of genes such as ENO2, HK2, and PFKFB3. However, the hypothesis is tested via only correlations in gene expressions, and intervention via c-Jun inhibitor SP600125. The authors need to test their hypothesis with more specific interventions, e.g., KO and KD interventions.

Response:

As the reviewer's comment was reasonable, we added the following results in Figure 5 and the following paragraph in the section of results.

Figure 5F and 5G.

It was further confirmed that c-Jun regulates the upregulation of ENO2, HK2, and PFKFB3 protein expression and the promotion of lactate production in CAF cells by compressive stress. c-Jun knockdown by shRNA decreased the compression-induced upregulation of ENO2, HK2, and PFKFB3 protein expression cells (Figure 5F) and the compression-induced promotion of lactate production in CAF cells (Figure 5G).

Comment 4-2.

The pharmacologic inhibition of c-jun results in lower JNK, c-jun, ENO2, HK2, PFKFB3 (Fig. 5D). But in some of these genes, they still show a dose-dependent response to compressive stress, and some don't. It would be useful to show the quantification of the western blot for a clearer results, and discuss the difference in dose-dependent response.

Response:

As the reviewer's comment was reasonable, we added the quantification of the Western blot band and discussion as follows.

Page 14, line 342 - 346

However, the expression of glycolysis-related genes was not dependent on the degree of compressive stress. A possible hypothesis for this phenomenon is that the responses to compressive stress is dynamic over time or different for each gene. For an example, the responses to transforming growth factor beta are diverse between cell types and environmental conditions²⁸.

Comment 5.

The human data are not very informative or supportive of the authors' hypothesis because it is not possible to know what levels of stress were present in the samples. The authors misunderstood the difference between stiffness (measured via SWE) and compressive stresses when they present the clinical data. They have classified patients based on shear wave elastography (SWE), a method that measures stiffness (rigidity) of the tissue. They, however, claim they have classified based on the high and low compressive stresses, which is not correct.

Response:

As the reviewer's comment, tissue classification by SWE was not suitable for our study. Therefore, we reanalyzed **"breast cancer patient tissues"** and **"a pre-existing clinical database"** based on the expression of collagen and hyaluronan. **"The background why we used the expression of collagen and hyaluronan for tissue classification"**, **"new results and discussion"**, and **"new tissue classification"** were as follows.

The background

Page 10, line 241 - page 11, line 253

To investigate that the findings in our in vitro model are observed in cancer patients, breast cancer patient tissues had to be classified into low- and high-compression groups. However, it was not possible to directly measure compressive stress from cancer patient tissues. According to previous studies, compressive stress is a major solid stress in tumor tissue and increases the epicenter and periphery of tumor¹⁵. Solid stress increases with tumor size¹⁶ and its accumulation is correlated with the expression of ECMs such as HAS2, HAS3, COL1A1, and COL3A1^{17, 18}. In the stiffened environment of tumor tissue, tumor growth results in compressive stress due to the resistance of hyaluronan¹⁹. Therefore, to establish a molecular basis to classify patient tissues into low- and high-

compression groups, we analyzed the Pearson's correlation between tumor size and ECM expression and found that tumor size is significantly and positively correlated with the expression of HAS2, HAS3, and COL1A1 genes (Supplementary Figure S3). Thus, breast patient tissues were classified into low-compression and high-compression group based on the expression of HAS2, HAS3, or COL1A1 genes.

Supplementary Figure S3.

The reanalysis of breast cancer patient tissues

Page 11, line 253 – page 11, line 269

The expression of PFKFB3 gene was positively correlated with EMT and angiogenesis-related gene expression and metastasis size in the breast cancer patient tissues with high compressive stress.

Among the compression-upregulated metabolic genes, the expression of HK2 and PFKFB3 genes were significantly and positively correlated with that of HAS2, HAS3, and COL1A1 genes in breast cancer patient tissues (Figure 7A). There was no difference between gene expression-based groupings. The expression of ENO2, HK2, and PFKFB3 genes was significantly higher in the high-compression group than the low-compression one (Figure 7B). The expression of EMT- or angiogenesis-related genes was significantly upregulated in the high-compression group compared to the low-compression one. The expression of TWIST1, SNAI1, ZEB1, ZEB2, CDH1, CDH2, and MMP2 genes was higher

in the high-compression group than the low-compression one (Figure 7C). The expression of VEGFA and VEGFB genes was higher in the high-compression group than the low-compression one (Figure 7D). The expression of ENO2 gene was significantly and positively correlated with that of CDH1 and MMP2 genes. The expression of PFKFB3 gene was significantly and positively correlated with that of TWIST1, SNAI1, SNAI2, ZEB1, ZEB2, CDH1, CDH2, and MMP2 genes (Figure 7E). The expression of PFKFB3 gene was significantly and positively correlated with that of VEGFA and VEGFB genes (Figure 7E). Invasion size and positive lymph node number were increased, but not significantly, in the high-compression group compared to the low-compression one (Figure 7G and 7H). Metastasis size was significantly increased in the high-compression group compared to the low-compression one (Figure 7I).

The reanalysis of a pre-existing clinical database

Page 12, line 283 – page 13, line 320

Next, we analyzed the correlation of compression-upregulated metabolic genes with compressive stress markers (COL1A1, COL3A1, HAS2, and HAS3 genes) and tumor size in breast cancer subtypes: luminal A (LumA), luminal B (LumB), Her2, and triple negative (TN). In Figure C, PFKFB3 gene was significantly and positively correlated with HAS2 gene in LumA. ENO2 gene was significantly and positively correlated with HAS3 gene, whereas PFKFB3 gene was significantly and positively correlated with COLA1, COL3A1, and HAS genes in LumB. ENO2 gene was significantly and positively correlated with COL1A1 gene, whereas PFKFB3 gene was significantly and positively correlated with COL1A1, COL3A1, HAS2, and HAS3 in Her2. HK2 gene was significantly and positively correlated with tumor size, whereas PFKFB3 was significantly and positively correlated with COL3A1, HAS2, and HAS3 genes in TN. Since PFKFB3 gene is significantly and positively correlated with compressive stress markers in all subtypes, its correlation with EMT- or angiogenesis-related genes was further analyzed. In Figure 8D, among the EMT-related genes, SNAI2, ZEB1, and

ZEB2 genes were significantly and positively correlated with PFKFB3 gene in LumA. SNAI2, ZEB1, MMP2 genes were significantly and positively correlated with PFKFB3 gene in LumB. SNAI2, ZEB1, ZEB2, CDH1, CDH2, and MMP2 genes were significantly and positively correlated with PFKFB3 gene in Her2. TWIST1, SNAI2, ZEB1, ZEB2, and MMP2 genes were significantly and positively correlated with PFKFB3 gene in TN. In Figure 8E, the expression of ZEB2 gene was significantly upregulated in the PFKFB3-high group of LumA compared to the PFKFB3-low one. The expression of MMP2 gene was significantly upregulated in the PFKFB3-high group of LumB compared to the PFKFB3-low one. The expression of SNAI2, ZEB1, ZEB2, CDH2, and MMP2 genes was significantly upregulated in the PFKFB3-high group of Her2 compared to the PFKFB3-low one. The expression of TWIST1, SNAI2, ZEB1, ZEB2, and MMP2 genes was significantly upregulated in the PFKFB3-high group of TN compared to the PFKFB3-low one. In Figure 8F, among angiogenesis-related genes, VEGFA and VEGFB genes were significantly and positively correlated with PFKFB3 gene in LumA. VEGFB gene was significantly and positively correlated with PFKFB3 gene in LumB. VEGFA and VEGFB genes were significantly and positively correlated with PFKFB3 gene in Her2. VEGFA, VEGFB, and VEGFD genes were significantly and positively correlated with PFKFB3 gene in TN. In Figure 8G, the expression of VEGFB and VEGFD genes was significantly upregulated in the PFKFB3-high group of LumA compared to the PFKFB3-low one. The expression of VEGFB gene was significantly upregulated in the PFKFB3-high group of LumB compared to the PFKFB3-low one. The expression of VEGFA gene was significantly upregulated in the PFKFB3-high group of Her2 compared to the PFKFB3-low one. The expression of VEGFA, VEGFB, and VEGFD gene was significantly upregulated in the PFKFB3-high group of TN compared to the PFKFB3-low one. Based on the correlation and expression analyses, PFKFB3 expression was associated with tumor progression in all breast subtypes. Therefore, we investigated whether PFKFB3 expression is related with a poor prognosis of breast cancer patients. In Figure 8H, PFKFB3 expression is significantly associated with the a poor prognosis of breast cancer patients.

Discussion

Page 15, line 363 – page 16, line 383

The degree of compressive stress within tumor tissues may be exploited as an index for cancer diagnosis. Unfortunately, however, it is not possible to directly measure solid stresses including compression from cancer patient tissues. Therefore, based on the expression of compressive stress markers (collagens and hyaluronan), we classified breast cancer patient tissues and a pre-existing clinical database into low- and high-compression group. In our study, the expression of the compression-upregulated metabolic genes, the EMT-related genes, and angiogenesis-related genes was higher in the high-compression group of breast cancer patient tissues than low-compression one. In addition, metastasis size was significantly bigger in the high-compression group than low-compression one. Especially, of the compression-upregulated metabolic genes, PFKFB3 gene was significantly and positively correlated with EMT- and angiogenesis-related genes as well as compressive stress markers. Since the outcomes from biochemical and mechanical signal transduction are both contained in solid tumor tissues ²⁰, we further analyzed cancer database to investigate whether our findings are observed in a large pre-existing clinical samples. Among the compression-upregulated metabolic genes, the expression of PFKFB3 gene was significantly and positively correlated with that of compressive stress markers (COL3A1, HAS2, and HAS3) in the LumB, Her2, and TN types of a clinical database (BRCA metabric). The expression of EMT- and angiogenesis-related factors were positively significantly correlated with that of PFKFB3 and higher in PFKFB3-high group than PFKFB3-low one. Breast cancer patients with high PFKFB3 expression showed a significantly a poorer prognosis than those with low PFKFB3 expression. Our analysis of the cancer database may show the possibility that the high compressive stress state of patient tissues can be an index for cancer progression.

Tissue classification

Page 2, line 30 - 34 in Supplementary Information

Classification of patient tissues according to compressive stress

Compressive stress is a major solid stress in tumor tissue and increases both the epicenter and periphery of tumor ². Solid stress is positively correlated with the expression of ECMs such as HAS2, HAS3, COL1A1, and COL3A1 ^{3,4}. Tumor growth results in compressive stress due to the resistance of hyaluronan ⁵. Therefore, breast cancer patient tissues were relatively classified into low- and high-compression groups based on the z-score of collagen and hyaluronan expression.

Comment 6.

In Fig 7, it is not possible to know how many CAFs and how many cancer cells contributed to the signal; therefore, the results are difficult to interpret in light of the rest of the study.

Response:

Unlike an in vitro model (to mimic a tissue environment), tumor growth-induced stress in tissue arises due to the microstructural and biochemical interaction between cancer and its stroma (Proc Natl Acad Sci U S A. 2012 Sep 18;109(38):15101-8). Therefore, it may be important to understand the integrated outcome from a tissue rather than the individual outcomes from cancer or CAF cells. In our tissue analysis, the expression of compression-upregulated metabolic genes (especially, PFKFB3) was significantly positively correlated with compressive stress markers (collagens and hyaluronan) and associated with the upregulation of tumor progression-related genes. These results were able to be interpreted in the other parts of our study.

Comment 7.

the authors need to present the western blots for shRNA KD to show the quality of KD.

Response:

As the reviewer's comment was fairly reasonable, we added the western blots for shRNA KD in Figure 4E.

Figure 4E

Comment 8.

The authors also need to have more controlled experiments: the group with no compression but treated with SP600125 is missing in Fig. 5E.

Response:

Since the reviewer's comment was reasonable, we included the group with no compression but treated with SP600125 in Figure5E.

Comment 9.

The frequent use of broken y axes is overused, confusing and unnecessary in most cases.

Response:

As the reviewer's comment, all broken y axes were revised as follows.

Figure 4C-F

Figure 4E

We hope that you will be pleased with this revision and finally recognize how useful our finding is.

REVIEWERS' COMMENTS:

Reviewer #2 (Remarks to the Author):

The authors have addressed most of my comments, and enhanced the ms with more data. There is one comment about the compression of the alginate gel that their answer is not very convincing, but it would be a difficult comment to address. Hence, the ms is acceptable for publication.